# TacoPrompt: A Collaborative Multi-Task Prompt Learning Method for Self-Supervised Taxonomy Completion

**Hongyuan Xu, Ciyi Liu, Yuhang Niu, Yunong Chen,**
**Xiangrui Cai, Yanlong Wen,*Xiaojie Yuan**
College of Computer Science, Nankai University, Tianjin, China
{xuhongyuan, liuciyi, niuyuhang, chenyunong}@dbis.nankai.edu.cn
{caixr, wenyl, yuanxj}@nankai.edu.cn

## Abstract

Automatic taxonomy completion aims to attach the emerging concept to an appropriate pair of hypernym and hyponym in the existing taxonomy. Existing methods suffer from the overfitting to leaf-only problem caused by imbalanced leaf and non-leaf samples when training the newly initialized classification head. Besides, they only leverage subtasks, namely attaching the concept to its hypernym or hyponym, as auxiliary supervision for representation learning yet neglect the effects of subtask results on the final prediction. To address the aforementioned limitations, we propose TacoPrompt, a **Co**llaborative Multi-**Ta**sk Prompt Learning Method for Self-Supervised **Ta**xonomy **Co**mpletion. First, we perform triplet semantic matching using the prompt learning paradigm to effectively learn non-leaf attachment ability from imbalanced training samples. Second, we design the result context to relate the final prediction to the subtask results by a contextual approach, enhancing prompt-based multi-task learning. Third, we leverage a two-stage retrieval and re-ranking approach to improve the inference efficiency. Experimental results on three datasets show that TacoPrompt achieves state-of-the-art taxonomy completion performance. Codes are available at https://github.com/cyclexu/TacoPrompt.

## 1 Introduction

A taxonomy is a tree-like hierarchical structure organized around hypernym-hyponym ("*is-a*") relations between concepts. It has been widely used in downstream natural language processing (NLP) tasks such as named entity recognition (Wang et al., 2021b), transfer learning (Luo et al., 2022) and language models (Bai et al., 2022).

Most of the existing taxonomies are maintained by domain experts manually, which is labour-intensive and time-consuming. As new concepts

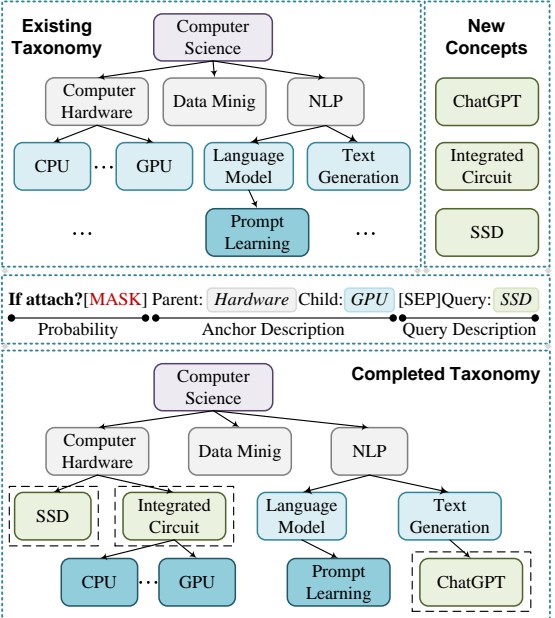

Figure 1: An example of attaching new concepts to the existing "Computer Science" taxonomy by triplet semantic matching using prompt learning paradigm.

continuously emerge, it is infeasible to manage the overwhelming amount of new content in online streaming (Zhang et al., 2021). To this end, considerable effort has been devoted to automatic *taxonomy expansion task* (Shen et al., 2020; Yu et al., 2020; Wang et al., 2021a), where a new concept (*query*) is only attached to the most appropriate hypernym (*parent*) as a leaf node while ignoring its potential hyponym (*child*) in the existing taxonomy. However, such a "leaf-only" assumption is inappropriate (Zhang et al., 2021) and results in large limitations (Wang et al., 2022) in real-world applications. Therefore, Zhang et al. (2021) proposes the *taxonomy completion task*, which aims to find an appropriate hypernym and hyponym pair (*anchor* or *position*) for the new concept, to satisfy the non-leaf attachment requirement. For example, the query "Integrated Circuit" is attached to the parent "Computer Hardware" and the child "GPU"

---
*Corresponding author.

instead of just to the parent as shown in Figure 1.

Several recent studies have achieved promising progress in taxonomy completion. Typically, they first leverage semantic (Wang et al., 2022; Arous et al., 2023) or structural (Jiang et al., 2022) information to generate the concept and the position representation using multiple tasks as supervision (Zhang et al., 2021). Then, these methods utilize the matching module to identify the attachment prediction for the given query based on the generated representation. Although achieving remarkable performance, these representation-based methods have two main limitations. First, they primarily leverage a newly initialized classification head as the matching module, whose ability overfits to leaf-only, i.e., the taxonomy completion task degrades to the expansion task (Wang et al., 2022), when learning from imbalanced leaf and non-leaf training samples. Second, these methods only leverage subtasks, namely attaching the concept to its hypernym or hyponym, as auxiliary supervision for representation learning (Zhang et al., 2021; Wang et al., 2022; Jiang et al., 2022) while neglecting the results of subtasks, which are proven effective in the final prediction (Wei et al., 2022).

To effectively address the aforementioned limitations, we propose TacoPrompt for self-supervised taxonomy completion in this paper. First, we leverage the pre-trained language model (PLM) by the prompt learning paradigm to perform triplet semantic matching for the hierarchical "*is-a*" relations prediction between the query, candidate hypernym and hyponym concepts, as illustrated in Figure 1. Prompt learning paradigm (Schick and Schütze, 2021; Liu et al., 2023a) reuses the PLM's masked language modelling (MLM) head instead of introducing the randomly initialized classification head, requiring fewer training samples to learn the target ability. Thus, we leverage it to address the overfitting problem caused by imbalanced training samples. Second, we extend the multi-task learning from representation-based methods to the cross-encoder-based framework by taking advantage of prompt learning's effectiveness in multi-task learning scenarios (Fu et al., 2022; Sanh et al., 2022). Moreover, we design the result context to relate the final prediction to the subtask results by a contextual approach to enhance prompt-based multi-task learning. Specifically, we design the hard result context using explicit answer tokens and the soft result context using learnable hidden vectors. Third,

we leverage a two-stage retrieval and re-ranking approach to reduce the cross-encoder's expensive inference computation costs, increasing the inference efficiency by up to 1,000 times.

We conduct extensive experiments on three taxonomy datasets to compare TacoPrompt with state-of-the-art taxonomy expansion and completion methods. The comparative results show that Taco-Prompt outperforms previous state-of-the-art methods by a large margin, such as by 11.1% in hit@1 and 9.1% in recall@10 on average.

Our contributions can be summarized as follows:

- We propose TacoPrompt, a self-supervised taxonomy completion framework that leverages the cross-encoder by the prompt learning paradigm to effectively learn from imbalanced training samples and multiple tasks.

- We design the hard and soft result context to relate the final prediction to the results of subtasks by a contextual approach to enhance prompt-based multi-task learning.

- We leverage a retrieval and re-ranking approach to improve the inference efficiency. Experimental results on three taxonomy datasets show that TacoPrompt outperforms previous methods by a large margin in the taxonomy completion task.

## 2 Related Work

### 2.1 Taxonomy Expansion and Completion

Numerous investigations have been conducted on automatically updating the existing taxonomies. Based on the assumption of the problem, the studies can be divided into two main categories: taxonomy expansion and completion. In taxonomy expansion, researchers (Shen et al., 2020; Yu et al., 2020; Ma et al., 2021; Wang et al., 2021a; Liu et al., 2021; Takeoka et al., 2021; Zhai et al., 2023; Jiang et al., 2023; Cheng et al., 2022; Phukon et al., 2022) suggest that new concepts can only be added as leaf nodes. However, this assumption is inappropriate for real-world applications: the query nodes can also be added as non-leaf nodes to the taxonomy (Zhang et al., 2021). To address this issue, Zhang et al. (2021) defines a new task named taxonomy completion, where new concepts can be added at any position within the taxonomy and proposes to use multiple score functions to find the most appropriate position for the new concept.

Wang et al. (2022) and Jiang et al. (2022) further develop it by incorporating important sibling relations to enhance the nodes' representations. Arous et al. (2023) observes that distant nodes can also improve the representation and proposes a position-enhanced node representation learning method. Besides, Zeng et al. (2021) generates potential concepts before completing the taxonomy. Xia et al. (2023) proposes a new two-stage completion problem to find the parent first and then all children under the fixed parent. Note that this newly defined problem is out of the scope of this paper.

We observe that all previous completion methods focus on better representation learning with semantic information (Wang et al., 2022; Arous et al., 2023) or structural information (Zeng et al., 2021; Jiang et al., 2022; Arous et al., 2023). In this paper, we develop a cross-encoder-based method to perform triplet semantic matching between concept descriptions along with a two-stage retrieval and re-ranking approach to improve the inference efficiency of the cross-encoder-based method.

## 2.2 Prompt Learning Paradigm

Traditionally, the fine-tuning paradigm is leveraged to adapt the PLM to downstream tasks. The paradigm typically decodes the last layer's hidden state of the special `[CLS]` token with a task-specific classification head (Liu et al., 2021; Devlin et al., 2019). However, training such a head from scratch is prone to the overfitting problem (Dong et al., 2023) and requires a large number of labelled training samples, making fine-tuning paradigm infeasible in few-shot scenarios (Liu et al., 2023a).

Prompt learning paradigm has gained considerable attention since the birth of GPT-3 (Brown et al., 2020; Chen et al., 2022). The paradigm fills the gap between the learning objectives of downstream tasks and the PLM by reformulating downstream tasks into the MLM task form (Liu et al., 2023a) utilizing hard prompts (Chen et al., 2022; Schick and Schütze, 2021) or soft prompts (Hambardzumyan et al., 2021; Liu et al., 2023b; Cai et al., 2022). Instead of training a newly initialized classification head, the paradigm reuses the MLM head, which is well-trained in the PLM's pre-training stage, as the decoder for downstream tasks. Prompt learning has achieved impressive performance in text-related tasks (Brown et al., 2020; Dong et al., 2023; Schick and Schütze, 2021), especially in few-shot scenarios (Cai et al., 2022; Zhang et al., 2022; Schick

and Schütze, 2021) and multi-task scenarios (Sanh et al., 2022; Sun et al., 2022; Asai et al., 2022; Wang et al., 2023).

In the taxonomy expansion task, Xu et al. (2022) leverages prompt learning as the generation technique to generate candidate hypernyms directly based on the query description. In contrast, we aim to leverage prompt learning to address the imbalanced training samples and better multi-task learning challenges in the taxonomy completion task. We study the utilization of logic-related tasks' results by a contextual approach to enhance prompt-based multi-task learning.

## 3 Preliminaries

### 3.1 Prompt Learning Pipeline

The core idea of prompt learning is to convert the downstream task into the MLM task form. Assuming that $\mathcal{M}, \mathcal{F}$ denote the PLM and the template function respectively. Given a sentence input $x = (x_0, x_1, \ldots, x_n)$, the template function $\mathcal{F}$ generates the task-specific template $\hat{x} = \mathcal{F}(x)$ as:

$$\mathcal{F}(x) = [\text{CLS}] \, \mathcal{L}(x) \, [\text{MASK}] \, \mathcal{R}(x) \, [\text{SEP}], \quad (1)$$

where $\mathcal{L}(x), \mathcal{R}(x)$ refer to the left and right task-specific context. Next, contextual hidden vectors over $\hat{x}$ are produced sequentially by $\mathcal{M}$'s transformer layers. The hidden vector of the $[\text{MASK}]$ token is $h_{[\text{MASK}]} \in \mathbb{R}^d$, where $d$ is the hidden dimension of $\mathcal{M}$. Then, the MLM head of $\mathcal{M}$ generates logits to calculate each word $v$'s probability being filled in $[\text{MASK}]$ position as $P_{\mathcal{M}}([\text{MASK}] = v \mid h_{[\text{MASK}]})$. Given a verbalizer $f : \mathcal{Y} \mapsto \mathcal{V}$ that defines the mapping from the label space $\mathcal{Y}$ to the label token space $\mathcal{V}$, the probability distribution over the label $y$ is formulated as:

$$P(y \mid x) = g(P_{\mathcal{M}}([\text{MASK}] = v \mid \mathcal{F}(x)), v \in \mathcal{V}_y), \quad (2)$$

where $\mathcal{V}_y$ represents the subset of $\mathcal{V}$ and $g$ refers to a function converting the probability of label words to that of label $y$.

### 3.2 Problem Formulation

In this section, we formally define the taxonomy, the taxonomy completion task and the multiple tasks studied in this paper.

**Taxonomy.** We follow Shen et al. (2020); Zhang et al. (2021) and define a taxonomy as a directed acyclic graph $\mathcal{T} = (\mathcal{N}, \mathcal{E})$, where each node $n \in \mathcal{N}$ represents a concept and each edge $\langle n_p, n_c \rangle \in \mathcal{E}$

indicates a hypernym-hyponym relation expressing that $n_p$ is the most specific concept that is more general than $n_c$. Following Wang et al. (2021a), we assume there exists a corpus resource $\mathcal{D}$, from which each concept $n$'s corresponding description $d_n$ can be retrieved.

**Problem Definition.** The input of the taxonomy completion task includes two parts: (1) an existing taxonomy $\mathcal{T}^0 = \left(\mathcal{N}^0, \mathcal{E}^0\right)$ and (2) a set of new concepts $\mathcal{C}$. The overall goal is to complete $\mathcal{T}^0$ by attaching each new concept $q \in \mathcal{C}$ to the most appropriate candidate position $a$. The valid candidate position is defined by Zhang et al. (2021) as a concept pair $a = \langle p, c\rangle$, where $p \in \mathcal{N}^0$ and $c$ is one of the descendants of $p$ in the existing taxonomy. Note that $p$ or $c$ could be a "pseudo node" acting as a placeholder to represent that $q$ is semantically the most general or specific concept for the existing taxonomy. The existing taxonomy is completed to $\mathcal{T}' = \left(\mathcal{N}^0 \cup \{q\}, \mathcal{E}^0 \cup \{\langle p, q\rangle, \langle q, c\rangle\}\right)$ after attaching $q$. Following Shen et al. (2020); Manzoor et al. (2020); Zhang et al. (2021), we assume the new concepts in $\mathcal{C}$ are independent of each other and reduce the taxonomy completion task into $|\mathcal{C}|$ independent attachment optimization problems:

$$a_i^* = \underset{a_i \in \mathcal{A}}{\arg\max} \log P\left(q_i \mid a_i, \Theta\right), \qquad (3)$$

where $\forall i \in \{1, 2, \ldots, |C|\}$, $\mathcal{A}$ and $\Theta$ represent the set of valid candidate positions and model parameters, respectively.

**Multiple Tasks.** Following Zhang et al. (2021), we focus on a final task denoted as $T_a$, which aims to attach the query $q$ to the anchor $a$, along with two related subtasks $T_p$ and $T_c$ that aims to attach $q$ to the parent $p$ and child $c$, respectively.

# 4 Methodology

In this section, we first introduce our triplet semantic matching pipeline using the prompt learning for taxonomy completion in §4.1. Then, we enhance it by collaborative multi-task learning via the contextual approach in §4.2. Furthermore, we discuss the efficient inference process based on a two-stage approach in §4.3 and describe the self-supervised training details in §4.4.

## 4.1 Triplet Semantic Matching

As a potential knowledge base, the PLM has been proven effective in hypernym-hyponym relations learning (Wang et al., 2022; Jiang et al., 2022;

Arous et al., 2023). However, prior taxonomy completion methods focus on leveraging the PLM to generate semantic representations (Wang et al., 2022; Arous et al., 2023) or structural representations (Jiang et al., 2022) for the query and candidate positions while neglecting the powerful contextual dependency capture ability of the PLM. Thus, we utilize the PLM in a cross-encoder manner to distinguish hierarchical semantics between concepts. Specifically, given an input $x = \langle q, p, c\rangle$ and their corresponding description $d_q, d_p, d_c \in \mathcal{D}$, we first construct the triplet semantic context as:

$$\mathcal{S}(x) = \langle\text{Parent}\rangle d_p \langle\text{Child}\rangle d_c\,[\texttt{SEP}]\,\langle\text{Query}\rangle d_q, \tag{4}$$

where we utilize delimiter tokens, e.g., "$\langle\text{Parent}\rangle$", to identify the different types of concept semantic context. Then, we leverage the template function $\mathcal{F}$ to generate the task-specific template $\hat{x} = \mathcal{F}(x)$ for the taxonomy completion task as:

$$\mathcal{F}(x) = [\texttt{CLS}]\,\text{All}\,[\texttt{MASK}]\,\mathcal{S}(x)\,[\texttt{SEP}]. \tag{5}$$

Finally, we perform the prompt learning pipeline in Section 3.1 to calculate the probability of the query concept $q$ attaching to the candidate position $a$ in Equation 3 as:

$$P_{\mathcal{M}}([\texttt{M}] = \text{yes}) - P_{\mathcal{M}}([\texttt{M}] = \text{no}) \mid h_{[\texttt{M}]}, \tag{6}$$

where $[\texttt{M}]$ is short for $[\texttt{MASK}]$ and we address the multi-classification problem through binary classification (Zhang et al., 2021; Shen et al., 2020). We denote the probability as the score $s_a(q, a)$.

## 4.2 Collaborative Multi-Task Learning

Multi-task learning has been a typical approach in previous concept representation-based taxonomy completion methods (Wang et al., 2022; Jiang et al., 2022) since subtasks have been proven effective as an inductive bias for representation learning (Zhang et al., 2021). In contrast, we introduce our cross-encoder-based collaborative multi-task learning method in this section.

Following prior representation-based methods, the most straightforward multi-task learning approach is co-training the final task with subtasks. Specifically, we utilize the same semantic matching pipeline in Section 4.1 for subtasks $T_p$ and $T_c$, except that the semantic context only includes "$\langle\text{Parent}\rangle d_p$" and "$\langle\text{Child}\rangle d_c$", respectively, and the delimiter token before "$[\texttt{MASK}]$" in Equation 5 is changed from "All" to "Parent" or "Child".

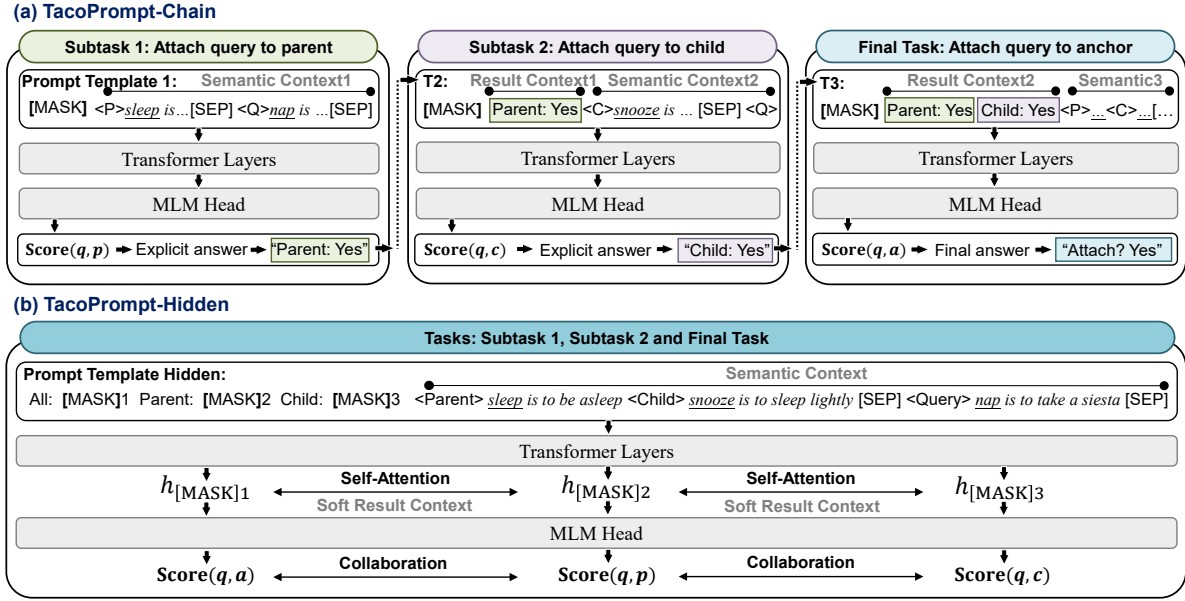

Figure 2: Illustration of our TacoPrompt framework.

To further enhance the prompt-based multi-task learning, we incorporate the subtask results as contexts into the template function defined in Equation 5, enabling the backbone LM to learn the dependency between the prediction for $T_a$ and the results of subtasks. Following this core idea, TacoPrompt has two variants named -Chain and -Hiden, that utilize the hard and the soft result context, respectively, as illustrated in Figure 2.

**TacoPrompt-Chain.** This variant utilizes explicit answers as the hard result context. Inspired by Chain-of-Thought Prompting (Wei et al., 2022), we address the task in a step-by-step approach. Specifically, we first utilize our semantic matching pipeline for the task $T_p$ for the corresponding result context denoted as $\mathcal{P}(x)$:

$$\mathcal{P}(x) = \begin{cases} \text{Parent}: \text{yes}, & \text{if } s_p(q, p) \geq 0, \\ \text{Parent}: \text{no}, & \text{otherwise}, \end{cases} \quad (7)$$

where $s_p(q, p)$ is the score of the query attaching to the parent calculated by Equation 6 in the $T_p$. Then, the template function of $T_c$ leverages the subtask result as the additional context:

$$\mathcal{F}_c(x) = \text{Child}\,[\text{MASK}]\,\mathcal{P}(x) \circ \mathcal{S}_c(x), \quad (8)$$

where $\circ$ is the string concatenation operation. We repeat this process for $T_c$ and generate the template function of $T_a$ as:

$$\mathcal{F}_a(x) = \text{All}\,[\text{MASK}]\,\mathcal{P}(x) \circ \mathcal{C}(x) \circ \mathcal{S}_a(x), \quad (9)$$

where $\mathcal{C}(x)$ is the result context of $T_c$. We leverage $s_a(q, a)$ as the final attachment score. The effects of different score function combinations will be discussed in Section 5.1.2.

**TacoPrompt-Hidden.** This variant utilizes hidden vectors as the soft result context. Specifically, we replace the hard result context $\mathcal{P}(x)$ and $\mathcal{C}(x)$ in Equation 9 to learnable "Parent: [MASK]" and "Child: [MASK]", respectively. TacoPrompt-Hidden utilizes multiple transformer layers with the self-attention mechanism to facilitate information interaction between hidden vectors of mask tokens, which are decoded for the result predictions of the corresponding tasks. Therefore, the results of subtasks are shown to the prediction of the final task through hidden vectors. Likewise, we also utilize $s_a(q, a)$ as the final score in this variant.

### 4.3 Efficient Inference

In this section, we analyze the inference time complexity of our method. We introduce a two-stage retrieval and re-ranking approach to improve the efficiency of the inference process.

**Inference Time Complexity.** Assuming that the number of queries is $|\mathcal{C}|$ and the number of candidate positions is $|\mathcal{A}|$. Our method utilizes the backbone LM to predict the probability for each candidate position, and the time complexity of the backbone LM is $\mathcal{O}(\Theta \times d \times l^2)$, where $\Theta$ and $d$ represent the parameter number and hidden dimension of the backbone LM, and $l$ is the average length of the

input sequence. Therefore, the inference time complexity of our method is $\mathcal{O}(|\mathcal{C}| \times |\mathcal{A}| \times \Theta \times d \times l^2)$. We observe that $|\mathcal{C}|$, $l$ are related to the task, and the choice of LM determines $d$, $\Theta$, the effects of which will be discussed in Section 5.1.3. Actually, the expensive computation, which is the main drawback of the cross-encoder-based method, mainly comes from the need to recompute $|\mathcal{A}|$ different triplets of sentences (Almagro et al., 2023).

**TacoPrompt++.** Following the retrieval and re-ranking approach in Information Retrieval (IR), we first leverage a lightweight representation-based method to recall the top 50 relevant candidate positions and then re-rank them using our method since the automatic taxonomy completion focuses more on top predictions. This two-stage approach improves the inference efficiency by $|\mathcal{A}|/100$ times since the representation in the first stage can be generated offline. More specifically, it is theoretically **146×, 858× and 1023× more efficient** on the SemEval-Food, MeSH and WordNet-Verb datasets. In this paper, we leverage the state-of-the-art representation-based method, TaxoComplete (Arous et al., 2023), as the retrieval stage method.

### 4.4 Self-supervised Training

In this section, we introduce the self-supervised training details of our framework.

**Self-supervision Generation.** Given one node $q \in \mathcal{N}^0$ in the existing taxonomy $\mathcal{T}^0 = \left(\mathcal{N}^0, \mathcal{E}^0\right)$ as query, we sample $1 + N$ anchors as the $q$'s anchor set $\mathbb{A}_q = \{\langle p^0, c^0 \rangle, \langle p^1, c^1 \rangle, \dots, \langle p^N, c^N \rangle\}$, including a positive anchor that $\langle p^0, q \rangle$ and $\langle q, c^0 \rangle$ are edges that exist in $\mathcal{E}^0$ and $N$ negative anchors of which at least one end does not exist in $\mathcal{E}^0$. We label each of these anchors by $\mathbf{y} = \{y_a, y_p, y_c\}$, where $y_p = I_{\mathcal{E}^0}\left(\langle p, q \rangle\right), y_c = I_{\mathcal{E}^0}\left(\langle q, c \rangle\right), y_a = y_p \wedge y_c$ and $I_{\mathcal{E}^0}(\cdot)$ is the indicator function. The label set of $q$ is denoted as $\mathbb{Y}_q = \{\mathbf{y}^0, \mathbf{y}^1, \dots, \mathbf{y}^N\}$. Finally, a training instance $X = \langle q, \mathbb{A}_q, \mathbb{Y}_q \rangle$ corresponds to the $q$ is created. By repeating the above process for each node $q \in \mathcal{N}^0$, we obtain the full training data $\mathbb{X} = \{X^0, X^1, \dots, X^{|\mathcal{N}^0|}\}$.

**Learning Objective.** We learn our model on $\mathbb{X}$ using the following objective:

$$\mathcal{L}(\Theta) = \lambda_a \mathcal{L}_a + \lambda_p \mathcal{L}_p + \lambda_c \mathcal{L}_c, \qquad (10)$$

where $\mathcal{L}_a$, $\mathcal{L}_p$, and $\mathcal{L}_c$ represent loss functions for different tasks, and we calculate them by the BCELoss. The hyper-parameters $\lambda_a$, $\lambda_p$ and $\lambda_c$ are weights to adjust the effect of each task.

## 5 Experiments

Detailed experimental settings, including the introduction of datasets, evaluation metrics and baselines, are described in Appendix A.

### 5.1 Experimental Results

#### 5.1.1 Comparison With Baselines.

The performance of all methods on SemEval-Food, MeSH, and WordNet-Verb datasets are summarized in Table 1. We discuss the questions below based on the comparative results.

**1. Which method achieves the most competitive performance in taxonomy completion?** Ours. As we can see in Table 1, both variants of our proposed method achieve statistically significantly better performance than the second-best method on three datasets. In particular, the two variants averagely increase MRR/H@1/R@10 by 0.134/16.6%/10.2%, 0.064/8.8%/8.2%, and 0.102/7.9%/8.4% over baselines on three datasets, respectively.

First, we observe that all methods that train a classification head from scratch perform relatively poorly in the non-leaf attachment scenario. Even the strongest baseline, TEMP, is defeated by Taxo-Complete, which leverages the cosine similarity for semantic matching, in attaching non-leaf queries on the MeSH and WordNet-Verb dataset. This illustrates that imbalanced leaf and non-leaf training samples will result in the overfitting to leaf-only problem of the newly initialized head. In contrast, our method leverages prompt learning to tune the pre-trained MLM head with imbalanced training samples, achieving advantageous performance in non-leaf attachment than other methods.

Besides, we notice that representation-based methods supervised with multiple tasks perform better than those not, as TMN, QEN, and TaxoEnrich outperform BERT+MLP, TaxoExpan and Arborist in most cases. On the other hand, we find that cross-encoder-based methods are more powerful than representation-based ones in the taxonomy completion, as TEMP has the best scores among baselines in 22 out of 24 columns in the total attachment scenario. Our improvement over TEMP confirms our intuition about prompt learning: it typically enables the LM to learn from multiple tasks' supervision, and we take advantage of the cross-encoder's contextual dependency capture ability to enhance multi-task learning via the result context. We will analyze more about the superiority of our prompt learning framework in Section 5.1.2.

| Datesets | Methods | Total | | | | | | | | Leaf | | | Non-leaf | | |
|---|---|---|---|---|---|---|---|---|---|---|---|---|---|---|---|
| | | MR↓ | MRR | R@1 | R@5 | R@10 | H@1 | H@5 | H@10 | MRR | H@5 | R@10 | MRR | H@5 | R@10 |
| SemEval-Food | BERT+MLP | 702.980 | 0.279 | 5.5 | 15.1 | 20.9 | 11.5 | 28.4 | 40.5 | 0.565 | 29.3 | 42.2 | 0.060 | 24.0 | 4.5 |
| | TaxoExpan | 371.291 | 0.286 | 5.7 | 13.3 | 18.0 | 11.5 | 26.4 | 34.5 | 0.477 | 30.1 | 35.6 | 0.130 | 8.0 | 3.6 |
| | Arborist | 256.491 | 0.290 | 13.0 | 18.0 | 21.0 | 26.4 | 34.5 | 38.5 | 0.466 | 39.0 | 38.5 | 0.146 | 12.0 | 6.7 |
| | TMN | 173.516 | 0.332 | 10.7 | 18.7 | 22.0 | 21.6 | 36.5 | 39.9 | 0.538 | 41.5 | 41.5 | 0.164 | 12.0 | 6.1 |
| | TaxoEnrich | 230.424 | 0.408 | 11.7 | 26.7 | 31.7 | 23.6 | 49.3 | 58.1 | 0.723 | 58.5 | 66.7 | 0.149 | 4.0 | 3.0 |
| | QEN | 336.554 | 0.439 | _21.9_ | 30.9 | 35.0 | _45.9_ | 58.8 | 64.9 | 0.732 | 64.2 | 68.9 | 0.209 | 32.0 | 9.1 |
| | TaxoComplete | 296.072 | 0.489 | 14.7 | 30.0 | 38.0 | 29.7 | 55.4 | 65.5 | 0.702 | 60.2 | 65.2 | 0.315 | 32.0 | 15.8 |
| | TEMP | _51.374_ | _0.579_ | 20.3 | _41.2_ | _47.9_ | 42.6 | _76.4_ | _81.1_ | _0.881_ | _81.3_ | _83.0_ | _0.348_ | _52.0_ | _21.0_ |
| | Ours-Chain | 51.643 | **0.717** | 28.6 | 51.4 | 60.8 | 60.1 | **85.8** | 89.2 | 0.886 | **87.8** | 86.7 | **0.587** | **76.0** | 40.9 |
| | Ours-Chain++ | - | - | 27.0 | 43.4 | 49.5 | 56.8 | 75.0 | 76.4 | - | 78.0 | 74.1 | - | 60.0 | 30.7 |
| | Ours-Hidden | **47.423** | 0.708 | **30.9** | 51.1 | 60.1 | **64.9** | **85.8** | 86.5 | **0.899** | **87.8** | 87.4 | 0.561 | **76.0** | 39.2 |
| | Ours-Hidden++ | - | - | 28.9 | 43.4 | 49.2 | 60.8 | 75.7 | 77.0 | - | 78.0 | 74.8 | - | 64.0 | 29.5 |
| MeSH | BERT+MLP | 9152.507 | 0.045 | 0.2 | 1.2 | 1.9 | 0.4 | 2.7 | 4.3 | 0.102 | 3.6 | 4.5 | 0.011 | 0.8 | 0.3 |
| | TaxoExpan | 1029.344 | 0.233 | 2.7 | 6.2 | 12.2 | 6.0 | 12.7 | 23.9 | 0.381 | 16.3 | 24.3 | 0.137 | 5.0 | 4.3 |
| | Arborist | 843.199 | 0.337 | 5.0 | 13.6 | 21.8 | 11.0 | 25.8 | 37.4 | 0.437 | 26.7 | 30.6 | 0.271 | 23.8 | 16.0 |
| | TMN | 567.831 | 0.372 | 7.2 | 17.3 | 24.6 | 15.9 | 33.6 | 43.8 | 0.525 | 38.4 | 40.7 | 0.271 | 23.4 | 14.1 |
| | TaxoEnrich | 393.062 | 0.424 | 7.4 | 22.4 | 31.0 | 16.2 | 42.6 | 52.5 | 0.619 | 51.3 | 54.1 | 0.296 | 24.1 | 15.9 |
| | QEN | 451.253 | 0.438 | 7.5 | 21.3 | 30.8 | 17.1 | 43.1 | 55.9 | 0.611 | 51.1 | 51.8 | 0.332 | 26.1 | 17.9 |
| | TaxoComplete | 357.494 | 0.540 | 10.8 | 29.3 | 41.1 | 24.5 | 54.1 | 63.9 | 0.605 | 53.8 | 52.5 | _0.500_ | _54.8_ | _34.1_ |
| | TEMP | _80.291_ | _0.612_ | _13.8_ | _35.3_ | _48.0_ | _31.4_ | _66.5_ | _77.5_ | _0.839_ | _75.4_ | _77.6_ | 0.471 | 47.5 | 29.8 |
| | Ours-Chain | 76.579 | **0.678** | 17.4 | 41.8 | **56.4** | 39.6 | 72.9 | 82.4 | 0.833 | 77.4 | 78.2 | **0.583** | 63.2 | 43.0 |
| | Ours-Chain++ | - | - | 17.9 | 42.1 | 53.7 | 40.8 | 71.2 | 76.4 | - | 71.9 | 66.1 | - | **69.7** | **46.1** |
| | Ours-Hidden | **49.140** | 0.674 | 17.9 | **42.4** | 55.9 | 40.7 | **74.6** | **84.6** | **0.868** | **79.0** | **81.5** | 0.554 | 65.1 | 40.2 |
| | Ours-Hidden++ | - | - | **18.3** | 40.8 | 53.0 | **41.5** | 71.8 | 78.5 | - | 75.4 | 68.9 | - | 64.0 | 43.2 |
| WordNet-Verb | BERT+MLP | 5858.466 | 0.113 | 2.6 | 5.7 | 7.6 | 4.0 | 8.8 | 11.7 | 0.202 | 10.1 | 13.7 | 0.015 | 3.6 | 1.0 |
| | TaxoExpan | 1752.271 | 0.215 | 4.1 | 11.4 | 15.1 | 6.1 | 17.1 | 22.5 | 0.354 | 20.5 | 26.7 | 0.057 | 3.1 | 1.7 |
| | Arborist | 1455.251 | 0.246 | 3.8 | 11.0 | 15.5 | 5.7 | 15.5 | 21.6 | 0.331 | 16.2 | 21.8 | 0.148 | 12.8 | 8.4 |
| | TMN | 1513.634 | 0.290 | 5.4 | 14.7 | 20.7 | 8.1 | 21.2 | 29.1 | 0.425 | 23.8 | 32.8 | 0.136 | 10.7 | 6.8 |
| | TaxoEnrich | 5462.075 | 0.179 | 3.9 | 9.0 | 12.3 | 5.8 | 13.6 | 18.4 | 0.313 | 16.8 | 22.6 | 0.025 | 0.5 | 0.4 |
| | QEN | 1730.755 | 0.404 | 9.1 | 23.3 | 31.0 | 13.9 | 34.0 | 43.9 | 0.568 | 38.6 | 48.4 | 0.224 | 15.3 | 11.8 |
| | TaxoComplete | 2661.488 | 0.407 | 9.0 | 22.2 | 30.9 | 13.6 | 31.7 | 40.8 | 0.487 | 32.7 | 41.3 | _0.315_ | _27.6_ | _19.1_ |
| | TEMP | _960.536_ | _0.450_ | _13.3_ | _30.6_ | _37.5_ | _20.3_ | _45.9_ | _55.0_ | _0.692_ | _53.4_ | _62.8_ | 0.182 | 15.3 | 9.5 |
| | Ours-Chain | 597.098 | 0.546 | 18.5 | 35.5 | 45.2 | 28.3 | 49.3 | 58.9 | 0.692 | 52.9 | 61.9 | **0.385** | 34.7 | 26.8 |
| | Ours-Chain++ | - | - | **19.7** | 36.2 | 42.9 | **30.2** | 48.8 | 53.4 | - | 50.2 | 54.4 | - | **42.9** | **30.2** |
| | Ours-Hidden | **436.799** | 0.557 | 18.3 | **36.9** | **46.5** | 28.0 | **52.3** | **62.5** | **0.762** | **56.5** | **65.8** | 0.370 | 35.2 | 25.3 |
| | Ours-Hidden++ | - | - | 17.2 | 35.4 | 42.1 | 26.3 | 48.5 | 53.1 | - | 50.9 | 54.5 | - | 38.8 | 28.4 |

Table 1: Overall results on all three datasets. All baseline methods are reproduced for non-leaf attachment results. The best results of our and baseline methods are in bold and underlined respectively for comparison.

| Datasets | Chain | Chain++ | Hidden | Hidden++ |
|---|---|---|---|---|
| SemEval-Food | 40.716s | 0.284s | 12.811s | 0.087s |
| MeSH | 241.971s | 0.278s | 81.145s | 0.094s |
| WordNet-Verb | 287.926s | 0.281s | 95.970s | 0.093s |

Table 2: Average inference time per query using a single NVIDIA RTX 3090 GPU on all three datasets.

**2. What is the performance comparison between the two variants?** They achieve overall comparable performance to each other. Specifically, the Hidden variant exhibits superior performance for leaf attachment, whereas the Chain variant excels in the non-leaf attachment scenario. This indicates that the hard result context provides better resistance to the overfitting problem than the learnable soft result context. For efficiency, Hidden is around 3x more efficient than Chain as shown in Table 2 since Chain has to perform cross-encoder-based semantic matching three times for all tasks. In contrast, Hidden tackles three tasks concurrently in

one semantic matching process.

**3. Is the two-stage retrieval and re-ranking approach effective in the inference stage?** Yes, TacoPrompt++ still achieves state-of-the-art performance while significantly decreasing the average inference time, as shown in Table 2, suggesting that the two-stage retrieval and re-ranking approach is feasible to improve the inference efficiency of the cross-encoder method in the taxonomy completion task. Interestingly, TacoPrompt++ even outperforms TacoPrompt in some cases. For example, the Chain++ increases H@1 by 1.9% on the Verb dataset. Therefore, developing a better method for the retrieval stage is expected in future studies due to the impressive effectiveness of the cross-encoder method in the taxonomy completion task.

### 5.1.2 Ablation Studies

Table 3 presents the ablation results under different settings. Assuming that $rc$, $mt$ and $pl$ represent the result context, multi-task learning and prompt

learning, respectively and the "-NL" short for the non-leaf scenario, we discuss the questions below.

**1. Is prompt learning effective in addressing the taxonomy completion?** Yes, methods w/t $pl$ outperform their counterparts w/o $pl$ under the same $rc$ and $mt$ settings on three datasets. For example, they averagely increase MRR-NL/R@10-NL on three datasets by 0.036/2.7% under the multi-task setting and 0.033/4.9% under the single-task setting. It indicates that leveraging the MLM head instead of a newly initialized classification head keeps the backbone LM from overfitting to the leaf attachment, the training sample number of which surpasses that of the non-leaf attachment as shown in Table 7. Besides, the improvement under the multi-task setting illustrates the effectiveness of prompt learning in learning from multiple tasks in the cross-encoder-based completion framework.

**2. Can the result context enhance prompt-based multi-task learning?** Yes, the performance of method w/o $rc$ drops by different extents, especially in the most important non-leaf attachment scenario in the taxonomy completion task (Wang et al., 2022). For example, R@10-NL drops 8.5%, 1.5% and 3.6% compared to the method w/t the hard result context on the Food, MeSH and Verb datasets, respectively. In addition, the total H@1 drops 2.7%, 2.6% and 0.6% compared to the method w/t the soft result context on three datasets. Together, the result context effectively relates the prediction of different tasks in a contextual approach and is proven effective in enhancing prompt-based multi-task learning.

Below, we discuss the results using different tasks and score function combinations on SemEval-Food, shown in Table 4.

**3. Which combo choices of task supervision and score functions perform better?** Methods supervised by all tasks $T_a$, $T_p$ and $T_c$ perform better. First, each task provides supervision for taxonomy completion as shown in lines 1-3. Second, co-training two or all tasks can introduce bias for better performance in our proposed cross-encoder-based framework, and methods leveraging all tasks during training in lines 6-7 achieve the overall best performance. Compared to utilizing the prediction probability of $T_a$ as the final score, combining that of $T_p$ and $T_c$ shows the potential to improve automatic completion performance since H@1 is further improved 2% as shown in line 7.

| Datasets | Settings | MRR | H@1 | R@10 | MRR-NL | R@10-NL |
|---|---|---|---|---|---|---|
| SemEval-Food | Ours-Chain | **0.717** | 60.1 | **60.8** | **0.587** | **40.9** |
| | Ours-Hidden | 0.708 | **64.9** | 60.1 | 0.561 | 39.2 |
| | w/o $rc$ | 0.678 | 62.2 | 57.2 | 0.498 | 32.4 |
| | w/o $rc$, $mt$ | 0.662 | 55.4 | 55.9 | 0.473 | 34.1 |
| | w/o $rc$, $pl$ | 0.651 | 53.4 | 55.0 | 0.441 | 28.4 |
| | w/o $rc$, $mt$, $pl$ | 0.643 | 48.6 | 54.0 | 0.443 | 28.4 |
| MeSH | Ours-Chain | **0.678** | 39.6 | **56.4** | **0.583** | **43.0** |
| | Ours-Hidden | 0.674 | 40.7 | 55.9 | 0.554 | 40.2 |
| | w/o $rc$ | 0.677 | 38.1 | 56.1 | 0.568 | 41.5 |
| | w/o $rc$, $mt$ | 0.654 | **41.8** | 54.1 | 0.526 | 37.6 |
| | w/o $rc$, $pl$ | 0.650 | 34.3 | 53.7 | 0.536 | 38.9 |
| | w/o $rc$, $mt$, $pl$ | 0.627 | 33.7 | 50.2 | 0.480 | 31.0 |
| WordNet-Verb | Ours-Chain | 0.546 | **28.3** | 45.2 | **0.385** | **26.8** |
| | Ours-Hidden | **0.557** | 28.0 | **46.5** | 0.370 | 25.3 |
| | w/o $rc$ | 0.536 | 27.4 | 44.2 | 0.357 | 23.2 |
| | w/o $rc$, $mt$ | 0.549 | 25.0 | 45.1 | 0.359 | 23.2 |
| | w/o $rc$, $pl$ | 0.539 | 24.9 | 44.6 | 0.337 | 21.8 |
| | w/o $rc$, $mt$, $pl$ | 0.552 | 26.1 | 45.8 | 0.337 | 20.9 |

Table 3: Ablation studies on all three datasets. Note that $rc$, $mt$ and $pl$ represent the result context, multi-task learning and prompt learning, respectively, and "NL" is short for the non-leaf scenario. In w/o $pl$ experiments, we utilize a two-layer MLP as the classification head to decode the hidden vector of the [CLS] token for attachment probability prediction.

| Ind. | $T_a$ | $T_p$ | $T_c$ | Score | MRR | H@1 | R@10-L | R@10-NL |
|---|---|---|---|---|---|---|---|---|
| 1 | ✓ | | | $s_a(q,a)$ | 0.662 | 55.4 | 87.4 | 34.1 |
| 2 | | ✓ | | $s_p(q,p)+s_p(c,q)$ | 0.651 | 58.1 | 75.6 | 37.5 |
| 3 | | | ✓ | $s_c(q,p)+s_c(c,q)$ | 0.681 | 55.4 | 84.4 | 38.6 |
| 4 | ✓ | ✓ | | $s_a(q,a)$ | 0.704 | 60.1 | 86.7 | 39.2 |
| 5 | ✓ | | ✓ | $s_a(q,a)$ | 0.685 | 58.1 | 88.9 | 36.9 |
| 6 | ✓ | ✓ | ✓ | $s_a(q,a)$ | 0.708 | 64.9 | 87.4 | 39.2 |
| 7 | ✓ | ✓ | ✓ | $s_p(q,p)+s_c(c,q)$ | 0.699 | 66.9 | 85.9 | 39.8 |
| 8 | ✓ | | ✓ | $s_p(q,p)+s_c(c,q)$ | 0.674 | 60.1 | 86.7 | 35.2 |

Table 4: Ablation studies of different task supervision and score function combinations on SemEval-Food. Note that the score function $s_i(x, y)$ refers to the probability of attaching $x$ to $y$ in the task $T_i$. "L" and "NL" are short for the leaf and non-leaf scenarios.

### 5.1.3 Further Discussions

We manually construct the pseudo semantic context by reversing descriptions of parent and child and the pseudo result context by replacing the hard result context with the ground truth or the fake truth. The inference performance of the trained model provided with different pseudo contexts on SemEval-Food is shown in Figure 3. We discuss the questions below.

**1. Does the language model predict the final attachment probability based on the result context?** Yes, our method's performance fluctuates with different pseudo hard result contexts. Specifically, the method performs better when the result context is closer to ground-truth and vice versa. For example, the method provided with all ground-truth result contexts beats that provided with all fake ones almost twice on MRR on the SemEval-Food dataset. This further illustrates that our method

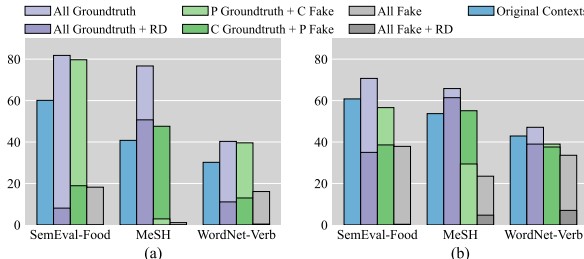

Figure 3: Inference performance with manually constructed pseudo semantic and result contexts. Note that "RD" is short for the "Reversed Description". We report the H@1 and R@10 results in (a) and (b).

| Variant | Backbone LM | Tr. Param. | MRR | H@1 | R@10 | R@10-NL | Avg. Inference Time |
|---------|-------------|------------|-----|-----|------|---------|---------------------|
| Hidden | BERT | 110M | 0.708 | 64.9 | 60.1 | 39.2 | 12.81s |
| | RoBERTa | 125M | 0.696 | 58.8 | 60.5 | 39.8 | 17.52s |
| | ALBERT | 125M | 0.707 | 57.4 | 60.1 | 42.6 | 20.36s |
| | ELECTRA | 14M | 0.687 | 59.5 | 59.2 | 37.5 | 9.55s |
| | DistilBERT | 66M | 0.681 | 52.7 | 56.6 | 34.7 | 11.16s |
| Chain | BERT | 110M | 0.717 | 60.1 | 60.8 | 40.9 | 40.72s |
| | RoBERTa | 125M | 0.716 | 60.1 | 60.5 | 40.9 | 49.47s |
| | ALBERT | 125M | 0.713 | 59.5 | 58.5 | 39.2 | 59.98s |
| | ELECTRA | 14M | 0.657 | 49.3 | 55.3 | 39.2 | 20.67s |
| | DistilBERT | 66M | 0.680 | 50.7 | 55.0 | 31.2 | 30.21s |

Table 5: Completion performance comparison of different backbone LMs on SemEval-Food. Tr. Param. refers to the number of trainable parameters. The average inference time is tested under the maximum batch size.

| Dataset | Backbone LM | R@1 | R@5 | R@10 | H@1 | H@5 | H@10 |
|---------|-------------|-----|-----|------|-----|-----|------|
| SemEval-Food | Llama-7B | 1.6 | 5.8 | 12.9 | 3.4 | 10.8 | 22.3 |
| | Llama-2-7B | 1.3 | 4.2 | 14.1 | 2.7 | 7.4 | 21.6 |
| | Ours-Chain++ | 27.0 | 45.3 | 53.1 | 56.8 | 76.4 | 77.7 |
| MeSH | Llama-7B | 0.6 | 5.4 | 12.0 | 1.5 | 11.6 | 22.5 |
| | Llama-2-7B | 0.5 | 5.7 | 12.1 | 3.4 | 12.1 | 21.9 |
| | Ours-Chain++ | 27.0 | 45.3 | 53.1 | 56.8 | 76.4 | 77.7 |
| WordNet-Verb | Llama-7B | 0.6 | 4.1 | 9.6 | 0.9 | 6.2 | 13.5 |
| | Llama-2-7B | 0.9 | 5.0 | 10.2 | 1.4 | 7.3 | 13.9 |
| | Ours-Chain++ | 27.0 | 45.3 | 53.1 | 56.8 | 76.4 | 77.7 |

Table 6: Completion performance of few-shot prompting the LLM with hard result context. We compare the performance using the two-stage retrieval and re-ranking approach due to the LLM's expensive inference computation costs.

relates the subtask results to the final prediction.

**2. What is the relationship between semantic and result context?** They provide mutual complementary information. First, the method provided with all fake result contexts can still predict correct final results, suggesting that it is robust to the noise results provided by subtasks. Conversely, the method provided with the reversed semantic context can be improved by ground-truth result contexts. Overall, our method can cross-consider semantic and result contexts for final prediction.

In addition, we discuss the effects of the backbone LM choice, namely the bi-directional LM as shown in Table 5 and the LLM as shown in Table 6. We discuss the questions below.

**3. Which backbone LM is the best choice to balance effectiveness and efficiency?** ELECTRA. In Table 5, we study the influence of backbone LMs, including RoBERTa (roberta-base) (Liu et al., 2019), ALBERT (albert-base-v2) (Lan et al., 2020), ELECTRA (google/electra-small-generator) (Clark et al., 2020) and DistilBERT (distilbert-base-uncased) (Sanh et al., 2019), on the taxonomy completion performance. The results show that the LMs with more parameters, namely ALBERT and RoBERTa, have yet to show obvious performance advantages over BERT. On the other hand, DistilBERT consistently performs poorly in attaching non-leaf queries. In contrast, ELECTRA achieves comparable results to BERT in the total and non-leaf scenarios with nearly one-eighth of parameters, indicating that it is the possible backbone LM choice for balancing effectiveness and efficiency in our framework.

**4. What is the performance of few-shot prompting LLMs with result context in taxonomy completion?** Poor. In Table 6, we utilize Llama-7B (Touvron et al., 2023a) and Llama-2-7B (Touvron et al., 2023b) to complete the taxonomy in the few-

shot prompting approach with hard result context. The implementation details are described in Appendix B.2. We can observe a large margin exists between few-shot prompting LLMs and our prompt learning framework under the same attachment probability prediction pipeline. We analyze the possible reason accounting for the LLM's poor performance is that they are unaware of the structural distribution of existing taxonomies. In contrast, our trainable framework is more feasible for the taxonomy completion.

## 6 Conclusion

In this paper, we propose TacoPrompt for self-supervised taxonomy completion. Specifically, we leverage the prompt learning paradigm to perform triplet semantic matching, addressing the overfitting to the leaf-only problem caused by imbalanced training samples. Then, we perform collaborative multi-task learning by designing the result context to enhance the prompt learning framework. Moreover, we propose a two-stage retrieval and re-ranking approach for efficient inference. Experimental results on three taxonomy datasets show that TacoPrompt outperforms the state-of-the-art methods by a large margin.

## Limitations

We consider that the proposed method has the following three limitations. (1) It has poor scalability to process long description texts. We truncate the long concept description to the fixed token length to satisfy the max sequence length limitation of the backbone LM. However, this approach may cause information distortion and result in performance degradation. (2) We ignore the sibling information, which is proven effective in taxonomy completion, in our framework. Potential future studies include incorporating more subtasks, e.g., finding the appropriate sibling concept for the query, in the proposed collaborative multi-task learning framework. (3) The proposed two variants perform taxonomy completion separately. We will explore their cooperation approach for better taxonomy completion performance in the future.

## Acknowledgements

We thank anonymous reviewers for their valuable comments. We also thank Yuxun Qu and Yuxiao Liu for their insightful suggestions. This research is supported by Chinese Scientific and Technical Innovation Project 2030 (No.2018AAA0102100), NSFC-General Technology Joint Fund for Basic Research (No.U1936206), National Natural Science Foundation of China (No.62077031, 62372252, 62002178).

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

| Dataset | $|\mathcal{N}|/|\mathcal{N}_{\text{trian}}|$ | $|\mathcal{E}|$ | #depth | #leaf rate | #candidates |
|---|---|---|---|---|---|
| SemEval-Food | 1486/1190 | 1,533 | 8 | 0.806 | 7313 |
| MeSH | 9710/8072 | 10,498 | 10 | 0.708 | 42970 |
| WordNet-Verb | 13936/11936 | 13,407 | 12 | 0.771 | 51159 |

Table 7: The dataset statistics. $|\mathcal{N}|$, $|\mathcal{E}|$ are the total number of nodes and edges. We count the proportion of leaf nodes in the train nodes and the number of candidate positions.

# A  Experimental Setup

## A.1  Datasets.

Following Wang et al. (2022), we evaluate Taco-Prompt on three taxonomy completion datasets. The **SemEval-Food** is a food taxonomy based on the largest taxonomy of SemEval-2016 Task 13 (Bordea et al., 2015). The Medical Subject Headings (**MeSH**) is a hierarchically organized vocabulary of medicine (Lipscomb, 2000). The **WordNet-Verb** based on WordNet 3.0 has fully developed taxonomy in SemEval-2016 Task 14 (Jurgens and Pilehvar, 2016). For each of these taxonomies $\mathcal{T} = (\mathcal{N}, \mathcal{E})$, we follow Wang et al. (2022) and divide $\mathcal{N}$ into non-overlapping train nodes $\mathcal{N}_{\text{train}}$, validation nodes $\mathcal{N}_{\text{validation}}$ and test nodes $\mathcal{N}_{\text{test}}$. Specifically, we randomly sample 1,000 nodes as validation nodes $\mathcal{N}_{\text{validation}}$ and another 1,000 nodes as test nodes $\mathcal{N}_{\text{test}}$ for WordNet-Verb, and we sample 10% nodes as validation nodes and other 10% nodes as test nodes for SemEval-Food and MeSH. The left nodes are leveraged as train nodes $\mathcal{N}_{\text{train}}$. Table 7 presents the statistical information of the three datasets.

## A.2  Evaluation Metrics.

Since the model outputs a ranking list of the candidate positions for each query concept, we utilize the following rank-based metrics for evaluation.

- **Mean Rank (MR)** calculates the macro average ranking of the query concept's ground-truth among all candidate positions. Smaller MR indicates better performance.

- **Mean Reciprocal Rank (MRR)** computes the macro average reciprocal rank of the true positions. Following Shen et al. (2020), we scale the original MRR by 10 to expand the performance gap between different models.

- **Recall**@$k$ counts the ratio of the true positions ranked in the top $k$ to the total true positions for all query concepts.

- **Hit@$k$** measures the number of query concepts with at least a ground-truth position within the top $k$ predictions, divided by the total number of query concepts.

It is worth noticing that the metric **Precision@$k$**, which is widely used in previous work (Zhang et al., 2021; Wang et al., 2022; Jiang et al., 2022), is proportional to **Recall@$k$**.

## A.3 Baseline Methods.

We compare our method with the following state-of-the-art taxonomy completion techniques.

- **TMN** (Zhang et al., 2021) proposes the taxonomy completion task. This framework leverages subtasks as auxiliary supervision signals for concept representation learning.

- **TaxoEnrich** (Jiang et al., 2022) leverages structural information via taxonomy contextualized embedding and taxonomy-aware sequential encoders. This method utilizes a query-aware sibling aggregator to augment position representations.

- **QEN** (Wang et al., 2022) generates semantic concept representation by pretrained language model. This framework evaluates sibling relations to reduce the pseudo-leaf noise.

- **TaxoComplete** (Arous et al., 2023) leverages semantic similarity through bi-encoders. This method utilizes direction-aware propagation to learn position-enhanced node representation.

Following Zhang et al. (2021), we adapt taxonomy expansion baselines, namely **BERT+MLP**, **TaxoExpan** (Shen et al., 2020) and **Arborist** (Manzoor et al., 2020), to the taxonomy completion task by concatenating the representation of the parent and the child as the representation of the corresponding candidate position. Note that all methods mentioned above focus on a better concept or candidate position representation. We adapt the state-of-the-art cross-encoder-based taxonomy expansion method **TEMP** (Liu et al., 2021) to the completion since our model is also based on the powerful cross-encoder.

## B Implementation Details

### B.1 Comparison with Baselines

We utilize BERT [1] (Devlin et al., 2019) as our backbone LM. For our method, we truncate the long concept description to 80 tokens to satisfy the max sequence length limitation of the backbone LM. We utilize the AdamW optimizer and set the learning rate to 3e-5, batch size to 6 and accumulation step to 2 for training the model. The hyperparameters $\mathcal{L}_a$, $\mathcal{L}_p$, $\mathcal{L}_c$ are set to 1.0 except $\mathcal{L}_c$ for MeSH -Hidden is set to 0.3. We sample 15 negative positions for each training instance. During the training process, we first train the model with self-supervised training data for 40 epochs and test it on the validation set. According to the early stopping strategy, the training process ends when the MRR score on the validation set does not increase within 5 epochs. We evaluate the best model on the test set. To fairly compare the proposed model with all baselines, we replace the backbone LM of **QEN** from distillBERT to BERT to align with our settings. We adapt the taxonomy expansion method, TEMP, to the taxonomy completion task by adding the candidate child concept to its taxonomy-path. All the experiments are accelerated by NVIDIA RTX 3090 GPU devices.

### B.2 Few-shot Prompting Llama

When few-shot prompting Llama, we construct examples depending on the bi-encoder-based semantic similarity. Specifically, we first leverage the query concept's most semantically similar concept in the training set to construct the example prompt by filling in ground-truth results of subtasks in Equation 9. Similarly, we add a negative example to the prompt. Then, we utilize Llama to perform the same inference pipeline of TacoPrompt-Chain, except that the result context is autoregressively generated, to calculate the final prediction score. We accelerate few-shot prompting Llama by a single A100 GPU device.

## C Effects of Delimiter Contexts

We have discussed the effects of the semantic context $\mathcal{S}(x)$ and the result context $\mathcal{P}(x)$ and $\mathcal{C}(x)$ in Section 5.1.3, but the effects of the delimiter context, e.g., "$\langle \text{Parent} \rangle d_p$" in the semantic context and "Parent: [MASK]" in the result context, are not exposed, which is essential in prompt learning (Liu

---

[1] https://huggingface.co/bert-base-uncased

| Context | Chain | | | Hidden | | |
|---|---|---|---|---|---|---|
| | MRR | H@1 | R@10 | MRR | H@1 | R@10 |
| Answers | 0.714 | 62.2 | 60.1 | 0.692 | 60.1 | 59.5 |
| Unmatch | 0.718 | 56.1 | 60.5 | 0.693 | 64.2 | 58.5 |
| None | 0.677 | 54.1 | 55.9 | 0.668 | 55.4 | 55.0 |
| Ours | 0.717 | 60.1 | 60.8 | 0.708 | 64.9 | 60.1 |

Table 8: Results of our method on SemEval-Food with different delimiter contexts in the semantic and the result context.

| Dataset | Backbone LM | R@1 | R@5 | R@10 | H@1 | H@5 | H@10 |
|---|---|---|---|---|---|---|---|
| SemEval-Food | Llama-7B | 1.6 | 5.8 | 12.9 | 3.4 | 10.8 | 22.3 |
| | Llama-2-7B | 1.3 | 4.2 | 14.1 | 2.7 | 7.4 | 21.6 |
| | Llama-2-70B-8Bit | 1.0 | 4.2 | 9.6 | 2.0 | 8.1 | 16.2 |
| | Ours-Chain++ | 27.0 | 45.3 | 53.1 | 56.8 | 76.4 | 77.7 |
| WordNet-Verb | Llama-7B | 0.6 | 4.1 | 9.6 | 0.9 | 6.2 | 13.5 |
| | Llama-2-7B | 0.9 | 5.0 | 10.2 | 1.4 | 7.3 | 13.9 |
| | Llama-2-70B-8Bit | 0.6 | 3.1 | 8.4 | 1.0 | 4.3 | 11.5 |
| | Ours-Chain++ | 27.0 | 45.3 | 53.1 | 56.8 | 76.4 | 77.7 |

Table 9: Performance of few-shot prompting the Llama in the taxonomy completion task.

et al., 2023a). In this section, we conduct experiments with different delimiter context designs.

In Table 8, "Answers" and "Unmatch" indicate that we replace the delimiter tokens in $\mathcal{P}(x)$ by "Parent Answer:" and "P:" respectively while keeping the delimiter tokens in semantic context unchanged as "⟨Parent⟩". We replace the delimiter tokens in $\mathcal{C}(x)$ in the same way. "None" represents that we remove delimiter tokens in both the semantic and result contexts.

Firstly, we observe that the delimiter context significantly helps the language model distinguish between different concepts. Without the delimiter context, our method experiences a substantial performance drop. Secondly, the best performance is achieved when the delimiter tokens preceding the [MASK] token align with those in the semantic context, such as both being set as "Parent". In contrast, the model's performance drops by different extents when the delimiter context in the result context does not match that in the semantic context ("Unmatch") or when it involves repetitive interference information ("Answers").

# D More Analysis of Performing LLMs

We have compared few-shot prompting Llama-7B and Llama-2-7B with our trainable framework in Section 5.1.3. We further study the taxonomy completion performance of larger Llama on SemEval-Food and WordNet-Verb as shown in Table 9. Surprisingly, the Llama-2-70B-8Bit consistently underperforms the smaller Llama in the taxonomy completion task. We analyze the possible reason behind this phenomenon, which is that LLM considers a larger concept set instead of understanding the specific distribution of the existing taxonomy. More specifically, LLMs may try to find the query's most specific hypernyms or hyponyms, which may not exist in the existing taxonomy.

Such observation can motivate further studies from two aspects. From one perspective, it is challenging to make LLMs understand the specific distribution of the existing taxonomy when reasoning the query's attachment answers. From another perspective, the LLM era urges taxonomy completion researchers to construct up-to-date datasets consisting of sufficiently new concepts for emergent concept attachment performance evaluation.