# OpenReview forum: "TacoPrompt: A Collaborative Multi-Task Prompt Learning Method for Self-Supervised Taxonomy Completion"
_EMNLP/2023/Conference — EMNLP 2023 Main_

### Official Review · Reviewer_Z6sr · 2023-08-05

**Soundness:** 3

**Excitement:**

3: Ambivalent: It has merits (e.g., it reports state-of-the-art results, the idea is nice), but there are key weaknesses (e.g., it describes incremental work), and it can significantly benefit from another round of revision. However, I won't object to accepting it if my co-reviewers champion it.

**Missing References:**

1.While there may not be any prior work specifically on prompt learning for taxonomy completion, I came across a relevant study titled "TEAM: A multitask learning based Taxonomy Expansion approach for Attach and Merge". This paper delves into multi-task scenarios in the domain of taxonomy expansion. Considering the similarities in the research context, I believe that the reference to this paper is missing in this paper.

**Paper Topic And Main Contributions:**

This paper proposed a frame work TacoPrompt, a prompt-based framework for self-supervised taxonomy completion. The taxonomy completion task involves attaching new concepts to the most suitable positions in an existing taxonomy.

The paper highlight the limitations of simply attaching new concepts to hypernyms without considering hyponyms. Instead, they emphasize the completion task, where new concepts are added as parent-child pairs in existing taxonomies.

The paper suggests incorporating two subtasks - hypernym and hyponym matching  to enhance concept representation learning. However, the existing methods lack a comprehensive understanding of how these subtasks contribute to the main task and ignore their interdependencies. Moreover, paper mentioned that difficulties arise in attaching non-leaf queries, especially in low-resource or imbalanced taxonomies. The paper aims to address these issues and provide insights into improving completion methods for taxonomies.

TacoPrompt utilizes two variants of result context, explicit answers and implicit hidden vectors, to facilitate task collaboration. Experimental results demonstrate that TacoPrompt outperforms state-of-the-art methods on benchmark taxonomy datasets. The paper also analyzes the effectiveness of different tasks, contexts, and backbone language models in the proposed framework. Overall, author claims that TacoPrompt offers a promising approach for efficient and accurate taxonomy completion in natural language processing tasks.

**Questions For The Authors:**

1. I am curious about why the framework performs poorly on the WordNet dataset compared to the other datasets.?

**Reasons To Accept:**

1. As claimed by author, The paper proposes a novel prompt-based framework, TacoPrompt, for the challenging task of taxonomy completion.  While the concept of attaching new concepts to hypernyms and considering hyponyms is not new and has been previously studied, the novelty of TacoPrompt lies in its application of prompt learning to tackle the taxonomy completion task. Prompt learning is a recent and innovative technique that converts the task into a masked language model (MLM) form, effectively transforming it into a fill-in-the-blank problem.

2. The authors conduct extensive experiments on three benchmark taxonomy completion datasets, comparing TacoPrompt with state-of-the-art methods and taxonomy expansion baselines. The comprehensive evaluation, along with ablation studies, demonstrates the effectiveness of the proposed approach in different scenarios and highlights its superior performance, especially for attaching non-leaf concepts.

**Reasons To Reject:**

1. The paper claims its novelty in utilizing prompt learning for taxonomy completion. However, it seems that the approach primarily revolves around applying a Masked Language Model (MLM) in a taxonomy sequence. This raises questions about the true novelty of the contribution.

2.  Moreover, The reported results do demonstrate improvements over baselines, including "TaxoExpan". However, considering that "TaxoExpan" already incorporates attention mechanisms and network information, it raises questions about the true significance of the proposed method's performance enhancement. It is worth noting that the current approach appears to lack the utilization of network information, which could be essential in the context of taxonomy expansion tasks. Additional clarification and analysis from the authors would be valuable to better comprehend the actual impact of their proposed approach, especially in comparison to methods that leverage network information.

3. The authors have introduced a relatively straightforward approach, but the presentation and explanation of the method tend to be  complex.

4. It seems that there are some inconsistencies and issues with the equations presented in the paper. For example... (1) What is the difference between PM ([MASK] and PM ([M]. (2). Equation 4 is difficult to understand.

**Reproducibility:**

4: Could mostly reproduce the results, but there may be some variation because of sample variance or minor variations in their interpretation of the protocol or method.

**Reviewer Confidence:**

3: Pretty sure, but there's a chance I missed something. Although I have a good feel for this area in general, I did not carefully check the paper's details, e.g., the math, experimental design, or novelty.

---

> ### Author Rebuttal · Authors · 2023-08-28
>
> Thank you for the thoughtful comments and valuable suggestions. Below, we provide our response to the questions and concerns.
>
> **Q1. The Novelty of Our Framework**
>
> A1. We have to emphasize that **reformulating the downstream task to a Masked Language Model (MLM) form is actually the core idea of the prompt learning paradigm [1]**. Please refer to the response to Reviewer tsdR#Q2 for more detailed discussions about the novelty of our framework.
>
> **Q2. The Lack of Network Information**
>
> A2. We agree with you that the network information contributes to the taxonomy expansion (TE) task and taxonomy completion (TC) task and many previous works have focused on utilizing such information [2, 3, 4]. **However, our work is not concentrated on better network information utilization**. In fact, we aim to address two issues that only exist in the TC task (and not in the TE task):
> (1) learning to attach non-leaf queries with limited training samples and (2) designing a better multi-task learning TC method.
> Our experiments have demonstrated that the prompt-based learning TC framework and collaborative multi-task learning using the result context can address the above two issues, which are the actual impacts of our proposed method.
>
> **Besides, we have compared many TC baselines that leverage network information**, including QEN and TaxoEnrich, as shown in Table 2 in the paper. **Moreover, we supplement the reproduction results of the latest published work TaxoComplete [7]**, which could be seen as contemporaneous work to our submission according to the Paper Submission Information of EMNLP 2023.
>
> TaxoComplete leverages the distance information in the network as self-supervision when performing TC. I hope these supplementary results can address your concerns and provide more comparisons. Note that '#' and '*' refer to leaf and non-leaf queries, respectively.
>
>
> | Food      | MR↓  | MRR  | R@1  | R@5  | R@10 | H@1  | H@5  | H@10 | MRR# | H#5  | R#10 | MRR* | H*5  | R*10 |
> | ------------ | ---- | ---- | ---- | ---- | ---- | ---- | ---- | ---- | ---- | ----  | ---- | ---- | ---- | ---- |
> | TaxoComplete | 96.039  | 0.507 | 13.9 | 30.0 | 38.3 | 28.4 | 60.1 | 71.6 | 0.764 | 65.6 | 70.9 | 0.322 | 34.6 | 14.8 |
> | Ours-Chain  | 51.643 | 0.717 | 28.6 | 51.4 | 60.8 | 60.1 | 85.8 | 89.2 | 0.886 | 87.8 | 86.7 | 0.587 | 76.0 | 40.9 |
> | Ours-Hidden | 47.423 | 0.708 | 30.9 | 51.1 | 60.1 | 64.9 | 85.8 | 86.5 | 0.899 | 87.8 | 87.4 | 0.561 | 76.0 | 39.2 |
>
> | MeSH     | MR↓  | MRR  | R@1  | R@5  | R@10 | H@1  | H@5  | H@10 | MRR# | H#5  | R#10 | MRR* | H*5  | R*10 |
> | ------------ | ---- | ---- | ---- | ---- | ---- | ---- | ---- | ---- | ---- | ---- | ---- | ---- | ---- | ---- |
> | TaxoComplete | 93.741 | 0.565 | 10.7 | 30.5 | 43.6 | 20.6 | 50.2 | 64.3 | 0.616 | 49.4 | 51.5 | 0.526 | 52.0 | 37.5 |
> | Ours-Chain |  76.579 | 0.678 | 17.4 | 41.8 | 56.4 | 39.6 | 72.9 | 82.4 | 0.833 | 77.4 | 78.2 | 0.583 | 63.2 | 43.0 |
> | Ours-Hidden |  49.140 | 0.674 | 17.9 | 42.4 | 55.9 | 40.7 | 74.6 | 84.6 | 0.868 | 79.0 | 81.5 | 0.554 | 65.1 | 40.2 |
>
> | Verb     | MR↓  | MRR  | R@1  | R@5  | R@10 | H@1  | H@5  | H@10 | MRR# | H#5  | R#10 | MRR* | H*5  | R*10 |
> | ------------ | ---- | ---- | ---- | ---- | ---- | ---- | ---- | ---- | ---- | ---- | ---- | ---- | ---- | ---- |
> | TaxoComplete | 596.347 | 0.424 | 7.7 | 21.1 | 30.4 | 12.2 | 31.6 | 42.6 | 0.529 | 33.2 | 44.4 | 0.319 | 25.4 | 16.4 |
> | Ours-Chain   | 597.098 | 0.546 | 18.5 | 35.5 | 45.2 | 28.3 | 49.3 | 58.9 | 0.692 | 52.9 | 61.9 | 0.385 | 34.7 | 26.8 |
> | Ours-Hidden  |  436.799 | 0.557 | 18.3 | 36.9 | 46.5 | 28.0 | 52.3 | 62.5 | 0.762 | 56.5 | 65.8 | 0.370 | 35.2 | 25.3 |
>
> **Q3. The Equation Problem**
>
> Q3.1 What is the difference between PM ([MASK] and PM ([M].
>
> A4.1 ‘[M]’ is short for ‘[MASK]’ and such an abbreviation is first denoted in line 234 in the paper.
>
> Q3.2 Equation 4 is difficult to understand.
>
> A3.2 Equation 4 indicates that the final prediction score is calculated by the difference between the probabilities of the ‘[MASK]’ token being predicted as ‘Yes’ and ‘No’.
>
> **Q4. The Performance on the WordNet-Verb Dataset**
>
> A4. Actually, not only our framework but also the previous SOTA method QEN performs relatively poorly on the WordNet-Verb dataset. Such relatively poor performance reflects the challenges of performing the TC task on this dataset.  We can conclude two main challenges of this dataset. First, the verb concept is semantically more abstract than the noun concept, making it harder to distinguish the hierarchical semantics of <parent, query, child>. Second, the WordNet-Verb dataset consists of several subtrees. The differences in the hierarchical distribution of these subtrees will increase the difficulty of distinguishing hierarchical semantics during training and inference.
>
> **Q5. Missing References**
>
> A5. TEAM is proposed to solve the TE task. We focus on introducing and comparing with the previous studies of the TC task instead of the TE task. Please refer to our response to Reviewer j9Kp#Q1 for more discussions about the difference between TC and TE.
>
> Thank you for the valuable suggestions. Please let us know if you have any other comments/questions.
>
> Best,
>
> Authors
>
> **References**
>
> [1] Chen X, Li L, Zhang N, et al. Decoupling knowledge from memorization: Retrieval-augmented prompt learning[J]. Advances in Neural Information Processing Systems, 2022, 35: 23908-23922.
>
> [2] Yu Y, Li Y, Shen J, et al. Steam: Self-supervised taxonomy expansion with mini-paths[C]//Proceedings of the 26th ACM SIGKDD International Conference on Knowledge Discovery & Data Mining. 2020: 1026-1035.
>
> [3] Liu Z, Xu H, Wen Y, et al. TEMP: taxonomy expansion with dynamic margin loss through taxonomy-paths[C]//Proceedings of the 2021 Conference on Empirical Methods in Natural Language Processing. 2021: 3854-3863.
>
> [4] Jiang M, Song X, Zhang J, et al. Taxoenrich: Self-supervised taxonomy completion via structure-semantic representations[C]//Proceedings of the ACM Web Conference 2022. 2022: 925-934.

---

### Official Review · Reviewer_6PQe · 2023-08-07

**Soundness:** 3

**Excitement:**

3: Ambivalent: It has merits (e.g., it reports state-of-the-art results, the idea is nice), but there are key weaknesses (e.g., it describes incremental work), and it can significantly benefit from another round of revision. However, I won't object to accepting it if my co-reviewers champion it.

**Missing References:**

Zeng, Qingkai, et al. "Enhancing taxonomy completion with concept generation via fusing relational representations." Proceedings of the 27th ACM SIGKDD Conference on Knowledge Discovery & Data Mining. 2021.
Hope to see some discussions and comparisons maybe with this paper since it's also one major baseline in other previous papers.

**Paper Topic And Main Contributions:**

The paper proposes TacoPrompt, which utilizes prompting and multi-task objectives to address the taxonomy completion task. It leverages the prompt-based template to compute the probability of the query concept being attached to the candidate position. And then the framework divides the learning objective into two subtasks to learn the relationships between query nodes and the candidate positions respectively. And the experiments show that the proposed method outperforms all compared baselines by a large margin.

**Questions For The Authors:**

A. Why is the current prompting template chosen in the first place? Would be good to see some ablation studies or analysis on the prompting template.

B. In some of the performance of the baselines reported in the main table, it seems that many methods performed worse than the baselines before them, which do not follow the patterns studied in the previous papers. Could you explain why these methods (TMN is better than TaxoEnrich and QEN in Food dataset, TMN is worse than Arborist in MeSH dataset, etc.) became worse in the comparisons?

C. Can you report the running time and the efficiency of the proposed method compared to other baselines? It seems to me that the framework is complex when the number of nodes becomes very large.

**Reasons To Accept:**

1. The proposed method of using prompts in the taxonomy completion task is interesting and the motivation is clear.
2. The writing is well-structured and easy to follow overall.
3. The experimental results show that the proposed framework is very effective.
4. The framework was extended to TacoPrompt-Chain and Hidden. These two variants are well-motivated and shown to be effective.

**Reasons To Reject:**

1. The overall novelty and contribution of the paper are limited. The prompting template is similar to what is described GenTaxo (Zeng et al., 2021) and TaxoEnrich (Jiang et al., 2022) where these two papers used pseudo sentences to generate the embeddings for the concepts. But the prompting template used in this paper is definitely more natural and effective. The multi-task learning objective is almost the same as what was used in TMN (Zhang et al., 2021), which also considers the relationships between query node and parent, query node and child respectively.
2. See the questions below.

**Reproducibility:**

3: Could reproduce the results with some difficulty. The settings of parameters are underspecified or subjectively determined; the training/evaluation data are not widely available.

**Reviewer Confidence:**

4: Quite sure. I tried to check the important points carefully. It's unlikely, though conceivable, that I missed something that should affect my ratings.

---

> ### Author Rebuttal · Authors · 2023-08-28
>
> Thank you for the thoughtful comments and valuable suggestions. Below, we provide our response to the questions and concerns.
>
> **Q1. The Novelty of Our Framework**
>
> A1.
> **GenTaxo and TaxoEnrich haven’t leveraged the prompt learning paradigm to solve the taxonomy completion (TC) task**. These methods focus on utilizing LMs to learn structural information of taxonomies by describing tree-like or path-like structures with natural language and feeding them into LMs for better structural representations, which is totally different from our main contributions that (1) leveraging prompt-based learning paradigm for better non-leaf attachment with limited labelled training samples; (2) designing the result context to improve the prompt-based multi-task learning for TC.
>
> It is also worth noting that **the multi-task learning objective is not claimed as the main contribution or novelty of our framework**. We focus more on the effectiveness of result context in realizing task collaboration to improve prompt-based multi-task learning for TC, which has not been discussed in previous TC methods.
>
> Please refer to the response to Reviewer tsdR#Q2 for more discussions about the novelty of our framework.
>
> **Q2. The Design Reasons and Ablation Studies of Our Prompt Template**
>
> A2. Our prompt template includes three different parts:
>
> 1\) delimiter context which aims to help the LM understand different sentences, for example, ''<Parent> sleep is to be asleep <Child>'' aims to help the LM understand that the description of the parent concept ''sleep'' is between ''<Parent>'' and ''<Child>'';
>
> 2\) task-specific context which aims to provide more semantical information, which refers to descriptions of the parent, child and query;
>
> 3\) result context that aims to relate the results of subtasks to the final prediction, which is one of main our contributions.
>
> We have performed ablation studies of 3\) in Table 3, discussed the effectiveness of 2\) and 3\) in Section 4.3 ''Effects of Different Contexts'' and studied the influence of different designs of 1\) in Appendix B ''Effects of Delimiter Context''.
>
> **Q3. The Performance of Baselines**
>
> Q3.1 Why does TMN perform better than TaxoEnrich on the Food dataset?
>
> A3.1 First, we found that all the results on the food dataset reported by QEN [3] are unconverged. Thus, we carefully tuned the parameters for each method and trained them to converge.
> The results show that TMN is better than TaxoEnrich (but not than QEN) on the Food dataset. Such results can be attributed to the sensitivity of TaxoEnrich to the scarcity of taxonomies. SemEval-Food is a low-resource dataset and its non-connectivity may cause the representation generation module of TaxoEnrich difficult to extract the structure information during the training.
>
> Second, description resources have been proven effective in providing extra semantic information in taxonomy-related tasks [1, 2, 3]. We leveraged descriptions to generate fasttext embeddings for TMN (it originally uses concepts’ surface names to generate embeddings), **aiming to reduce the impact of the use of extra semantic resources when comparing different methods.** The performance of TMN using different embedding generation resources is presented below. We can observe that **TaxoEnrich is better than TMN-Name, which follows the patterns studied in the previous papers.** Note that  '#' and '*' refer to leaf and non-leaf queries, respectively.
>
> | SemEval-Food | MR↓  | MRR  | R@1  | R@5  | R@10 | H@1  | H@5  | H@10 | MRR# | H#5  | R#10 | MRR* | H*5  | R*10 |
> | ------------ | ---- | ---- | ---- | ---- | ---- | ---- | ---- | ---- | ---- | ----  | ---- | ---- | ---- | ---- |
> | TMN-Desc  | 153.963 | 0.423 | 18.0 | 31.0 | 34.0 | 36.5 | 58.8 | 64.2 | 0.809 | 70.7 | 74.8 | 0.107 | 0.0 | 0.6 |
> | TaxoEnrich | 230.424 | 0.408 | 11.7 | 26.7 | 31.7 | 23.6 | 49.3 | 58.1 | 0.723 | 58.5 | 66.7 | 0.149 | 4.0 | 3.0 |
> | TMN-Name | 140.961 | 0.358 | 9.0 | 18.7 | 24.0 | 18.2 | 37.2 | 41.9 | 0.541 | 41.5 | 43.0 | 0.208 | 16.0 | 8.5 |
>
> Q3.2 Why does TMN perform worse than Arborist on the MeSH dataset?
>
> A3.2 TMN actually performs better than Arborist on the MeSH dataset as reported in Table 2 in the paper. We also experiment with surface name-based embeddings for TMN on the MeSH dataset and find that the performance of TMN is improved as shown in the table below. Such improvement can be due to the over-long average description length in MeSH, resulting in low-quality concept embeddings generated by averaging token embeddings.
>
> | MeSH | MR↓  | MRR  | R@1  | R@5  | R@10 | H@1  | H@5  | H@10 | MRR# | H#5  | R#10 | MRR* | H*5  | R*10 |
> | ------------ | ---- | ---- | ---- | ---- | ---- | ---- | ---- | ---- | ---- | ----  | ---- | ---- | ---- | ---- |
> | TMN-Desc  | 1342.775 | 0.195 | 2.7 | 7.4 | 11.0 | 6.0 | 15.5 | 23.1 | 0.350 | 19.7 | 22.7 | 0.094 | 6.5 | 3.4 |
> | Arborist | 1045.969 | 0.198 | 1.1 | 5.1 | 9.6 | 2.4 | 9.2 | 16.6 | 0.235 | 7.0 | 11.4 | 0.173 |13.8 |8.4 |
> | TMN-Name | 548.434 | 0.391 | 7.4 | 17.4 | 26.6 | 16.4 | 34.1 | 46.8 | 0.548 | 40.0 | 43.8 | 0.289 | 21.5 | 15.3 |
>
> Supplementary experiments illustrate that the performance of TMN is related to the representation generation resources, the effectiveness of which varies across datasets. Thank you for pointing out this issue and helping us to find that the description-based representation is not always the best for TMN.
>
> **Q4. The Efficiency of the Proposed Method**
>
> A4. Assuming that the number of queries is $|C|$ and the number of candidate positions is $|A|$, the inference time complexity of baselines is $O(|C|+|A|)$, while that of our method is $O(|C|*|A|)$. Such inference time complexity difference comes from the different base models (please refer to the response to Reviewer j9Kp#Q2 for more discussions about base models).
> The efficiency is indeed one of our method’s limitations as discussed in the paper. We will study how to improve efficiency in our future work.
>
> **Q5. Missing Reference: KDD 2021 GenTaxo**
>
> Q5.1 Why didn't we cite GenTaxo?
>
> A5.1 GenTaxo changes the problem assumption and proposes a new task, "taxonomy generation". It is different from the common definition of the TC task in that new concepts are generated instead of being given when completing the taxonomy. It focuses more on the new concept generation than on the TC.
>
> Q5.2 Why didn't we make GenTaxo our baseline?
>
> A5.2 As mentioned above, GenTaxo focuses on a different task instead of the TC task. Although the author claims that GenTaxo++ is suitable for the common TC task, its "completion" stage uses the existing TC method—TaxoExpan, which has been our baseline model.
> Besides, GenTaxo's output is unsuitable for our ranking-based metrics. GenTaxo generates the concept for the candidate position instead of a ranking list for potential parents.
>
> To provide more comparisons,  **we supplement the reproduction results of the latest published work TaxoComplete [7]**, which could be seen as contemporaneous work to our submission according to the Paper Submission Information of EMNLP 2023. Note that  '#' and '*' refer to leaf and non-leaf queries, respectively.
>
> | Food      | MR↓  | MRR  | R@1  | R@5  | R@10 | H@1  | H@5  | H@10 | MRR# | H#5  | R#10 | MRR* | H*5  | R*10 |
> | ------------ | ---- | ---- | ---- | ---- | ---- | ---- | ---- | ---- | ---- | ----  | ---- | ---- | ---- | ---- |
> | TaxoComplete | 96.039  | 0.507 | 13.9 | 30.0 | 38.3 | 28.4 | 60.1 | 71.6 | 0.764 | 65.6 | 70.9 | 0.322 | 34.6 | 14.8 |
> | Ours-Chain  | 51.643 | 0.717 | 28.6 | 51.4 | 60.8 | 60.1 | 85.8 | 89.2 | 0.886 | 87.8 | 86.7 | 0.587 | 76.0 | 40.9 |
> | Ours-Hidden | 47.423 | 0.708 | 30.9 | 51.1 | 60.1 | 64.9 | 85.8 | 86.5 | 0.899 | 87.8 | 87.4 | 0.561 | 76.0 | 39.2 |
>
> | MeSH     | MR↓  | MRR  | R@1  | R@5  | R@10 | H@1  | H@5  | H@10 | MRR# | H#5  | R#10 | MRR* | H*5  | R*10 |
> | ------------ | ---- | ---- | ---- | ---- | ---- | ---- | ---- | ---- | ---- | ---- | ---- | ---- | ---- | ---- |
> | TaxoComplete | 93.741 | 0.565 | 10.7 | 30.5 | 43.6 | 20.6 | 50.2 | 64.3 | 0.616 | 49.4 | 51.5 | 0.526 | 52.0 | 37.5 |
> | Ours-Chain |  76.579 | 0.678 | 17.4 | 41.8 | 56.4 | 39.6 | 72.9 | 82.4 | 0.833 | 77.4 | 78.2 | 0.583 | 63.2 | 43.0 |
> | Ours-Hidden |  49.140 | 0.674 | 17.9 | 42.4 | 55.9 | 40.7 | 74.6 | 84.6 | 0.868 | 79.0 | 81.5 | 0.554 | 65.1 | 40.2 |
>
> | Verb     | MR↓  | MRR  | R@1  | R@5  | R@10 | H@1  | H@5  | H@10 | MRR# | H#5  | R#10 | MRR* | H*5  | R*10 |
> | ------------ | ---- | ---- | ---- | ---- | ---- | ---- | ---- | ---- | ---- | ---- | ---- | ---- | ---- | ---- |
> | TaxoComplete | 596.347 | 0.424 | 7.7 | 21.1 | 30.4 | 12.2 | 31.6 | 42.6 | 0.529 | 33.2 | 44.4 | 0.319 | 25.4 | 16.4 |
> | Ours-Chain   | 597.098 | 0.546 | 18.5 | 35.5 | 45.2 | 28.3 | 49.3 | 58.9 | 0.692 | 52.9 | 61.9 | 0.385 | 34.7 | 26.8 |
> | Ours-Hidden  |  436.799 | 0.557 | 18.3 | 36.9 | 46.5 | 28.0 | 52.3 | 62.5 | 0.762 | 56.5 | 65.8 | 0.370 | 35.2 | 25.3 |
>
> Thank you for the valuable suggestions. Please let us know if you have any other comments/questions.
>
> Best,
>
> Authors
>
> **References**
>
> [1] Liu Z, Xu H, Wen Y, et al. TEMP: taxonomy expansion with dynamic margin loss through taxonomy-paths[C]//Proceedings of the 2021 Conference on Empirical Methods in Natural Language Processing. 2021: 3854-3863.
>
> [2] Wang S, Zhao R, Chen X, et al. Enquire one’s parent and child before decision: Fully exploit hierarchical structure for self-supervised taxonomy expansion[C]//Proceedings of the Web Conference 2021. 2021: 3291-3304.
>
> [3] Wang S, Zhao R, Zheng Y, et al. Qen: Applicable taxonomy completion via evaluating full taxonomic relations[C]//Proceedings of the ACM Web Conference 2022. 2022: 1008-1017.

---

### Official Review · Reviewer_tsdR · 2023-08-10

**Typos Grammar Style And Presentation Improvements:** None
**Soundness:** 4

**Excitement:**

4: Strong: This paper deepens the understanding of some phenomenon or lowers the barriers to an existing research direction.

**Missing References:**

[1] Liu Z, Xu H, Wen Y, et al. TEMP: taxonomy expansion with dynamic margin loss through taxonomy-paths[C]//Proceedings of the 2021 Conference on Empirical Methods in Natural Language Processing. 2021: 3854-3863.

[2] Xia F, Weng Y, He S, et al. Find Parent then Label Children: A Two-stage Taxonomy Completion Method with Pre-trained Language Model[C]//Proceedings of the 17th Conference of the European Chapter of the Association for Computational Linguistics. 2023: 1032-1042.

**Paper Topic And Main Contributions:**

This paper proposes a new method called TacoPrompt for taxonomy completion. Taxonomy completion aims to attach new concepts to appropriate hypernym-hyponym pairs in an existing taxonomy. The key ideas are: Formulate taxonomy completion as a masked language modeling task, which allows efficiently learning to attach non-leaf concepts with few labeled examples. Use a multi-task learning approach with two subtasks: attaching the concept to the hypernym, and attaching it to the hyponym. Introduce a contextual approach to integrate the subtask results, showing them as contexts to enable collaboration between tasks. Propose two variants: TacoPrompt-Chain uses explicit subtask answers as context, while TacoPrompt-Hidden uses implicit vector representations. Experiments on 3 datasets show significant improvements over prior state-of-the-art methods, especially for attaching non-leaf concepts.

**Questions For The Authors:**

A. The baseline methods used are not the most recent state-of-the-art. Could you compare against methods from [1] and [2] to better situate the improvements?

**Reasons To Accept:**

- Contributions are clearly articulated and convincingly demonstrated.
- Addresses limitations of prior work in non-leaf concept attachment.
- Strong empirical results, outperforming prior methods substantially.

---

Overall, the paper has solid experimentation, the results are relevant to the claims, and it demonstrates that using this method is reasonable. Therefore, I recommend a **Soundness=4.5**

**Reasons To Reject:**

- While the author claims to be the first to propose the MLM format for solving the taxonomy completion (TC) task, utilizing prompts to help BERT solve the TC task is not novel [1][2]. The author attributes the inspiration of TacoPrompt-Chain to Chain of Thought, but the author did not choose causal language models like GPT/OPT/Bloom/Llama to perform the TC task (to my knowledge, placing "Yes"/"No" at the end would allow a similar approach with TacoPrompt). We also do not know how well Large Language Models would perform.

---

Therefore, I still have doubts regarding Excitement, and if the author can adequately address my concerns, I am willing to reconsider the rating.

**Reproducibility:**

4: Could mostly reproduce the results, but there may be some variation because of sample variance or minor variations in their interpretation of the protocol or method.

**Reviewer Confidence:**

5: Positive that my evaluation is correct. I read the paper very carefully and I am very familiar with related work.

---

> ### Author Rebuttal · Authors · 2023-08-28
>
> We really appreciate your thoughtful comments and valuable suggestions. Below, we provide our response to the questions and concerns.
>
> **Q1. Missing Reference & The Choice of Baselines**
>
> A1. First, we actually have cited EMNLP 2021 TEMP and EACL 2023 ATTEMPT.
>
> Second, we want to clarify why we didn’t compare our method with TEMP or ATTEMPT on the taxonomy completion (TC) task.
>
> 1\) TEMP is designed for the taxonomy expansion (TE) task, whose input is limited to a query and a candidate parent, while the input of the TC task includes a candidate child additionally. That’s possibly why all TC studies (including the latest published TaxoComplete [7]) haven’t compared with TEMP on the TC task.
>
> 2\) ATTEMPT aims to solve a quite different task from the TC task. Specifically, it firstly finds the <parent, query> and secondly tries to find all children for the query by only considering the query's siblings under the fixed parent (instead of all candidate parents). ATTEMPT has neither published its code nor provided enough details on how to score all candidate <parent, child> pairs to perform the TC task.
>
> Additionally, **we supplement the reproduction results of the latest published work TaxoComplete [7]**, which could be seen as contemporaneous work to our submission according to the Paper Submission Information of EMNLP 2023. I hope these supplementary results can address your concerns about the choice of baselines and can provide more convincing comparisons. Note that  '#' and '*' refer to leaf and non-leaf queries, respectively.
>
>
> | Food      | MR↓  | MRR  | R@1  | R@5  | R@10 | H@1  | H@5  | H@10 | MRR# | H#5  | R#10 | MRR* | H*5  | R*10 |
> | ------------ | ---- | ---- | ---- | ---- | ---- | ---- | ---- | ---- | ---- | ----  | ---- | ---- | ---- | ---- |
> | TaxoComplete | 96.039  | 0.507 | 13.9 | 30.0 | 38.3 | 28.4 | 60.1 | 71.6 | 0.764 | 65.6 | 70.9 | 0.322 | 34.6 | 14.8 |
> | Ours-Chain  | 51.643 | 0.717 | 28.6 | 51.4 | 60.8 | 60.1 | 85.8 | 89.2 | 0.886 | 87.8 | 86.7 | 0.587 | 76.0 | 40.9 |
> | Ours-Hidden | 47.423 | 0.708 | 30.9 | 51.1 | 60.1 | 64.9 | 85.8 | 86.5 | 0.899 | 87.8 | 87.4 | 0.561 | 76.0 | 39.2 |
>
> | MeSH     | MR↓  | MRR  | R@1  | R@5  | R@10 | H@1  | H@5  | H@10 | MRR# | H#5  | R#10 | MRR* | H*5  | R*10 |
> | ------------ | ---- | ---- | ---- | ---- | ---- | ---- | ---- | ---- | ---- | ---- | ---- | ---- | ---- | ---- |
> | TaxoComplete | 93.741 | 0.565 | 10.7 | 30.5 | 43.6 | 20.6 | 50.2 | 64.3 | 0.616 | 49.4 | 51.5 | 0.526 | 52.0 | 37.5 |
> | Ours-Chain |  76.579 | 0.678 | 17.4 | 41.8 | 56.4 | 39.6 | 72.9 | 82.4 | 0.833 | 77.4 | 78.2 | 0.583 | 63.2 | 43.0 |
> | Ours-Hidden |  49.140 | 0.674 | 17.9 | 42.4 | 55.9 | 40.7 | 74.6 | 84.6 | 0.868 | 79.0 | 81.5 | 0.554 | 65.1 | 40.2 |
>
> | Verb     | MR↓  | MRR  | R@1  | R@5  | R@10 | H@1  | H@5  | H@10 | MRR# | H#5  | R#10 | MRR* | H*5  | R*10 |
> | ------------ | ---- | ---- | ---- | ---- | ---- | ---- | ---- | ---- | ---- | ---- | ---- | ---- | ---- | ---- |
> | TaxoComplete | 596.347 | 0.424 | 7.7 | 21.1 | 30.4 | 12.2 | 31.6 | 42.6 | 0.529 | 33.2 | 44.4 | 0.319 | 25.4 | 16.4 |
> | Ours-Chain   | 597.098 | 0.546 | 18.5 | 35.5 | 45.2 | 28.3 | 49.3 | 58.9 | 0.692 | 52.9 | 61.9 | 0.385 | 34.7 | 26.8 |
> | Ours-Hidden  |  436.799 | 0.557 | 18.3 | 36.9 | 46.5 | 28.0 | 52.3 | 62.5 | 0.762 | 56.5 | 65.8 | 0.370 | 35.2 | 25.3 |
>
> **Q2. The Novelty of Our Framework**
>
> A2. We've already explained why TEMP and ATTEMPT are not TC methods in the Q1 reply. It's also worth noting that **TEMP and ATTEMPT haven’t leveraged the prompt learning paradigm to solve TC tasks**. These methods focus on providing LMs with the structural information of taxonomies by describing path-like structures with natural language, which is totally different from our contributions.
>
> In this paper, we leverage the prompt-based learning paradigm to address two problems that exist only in TC (not in TE): (1) learning to attach non-leaf queries with limited training samples and (2) designing a better multi-task learning TC method.
>
> The core idea of the prompt learning paradigm is to reformulate the downstream tasks into the form of the [MASK] prediction to align with the PLM’s pre-training task [1, 2], which for our backbone LM is the MLM task [3].
> Such a paradigm has been proven effective in learning with limited labelled training samples [4]. At the same time, we observe that the TC task suffers from task degradation caused by an unbalanced training data distribution (few non-leaf training samples). Therefore, **our first contribution is to reformulate the TC task into the MLM form based on the core idea of the prompt learning paradigm, improving the performance of the non-leaf query attachment with limited training samples.**
>
> Besides, the prompt learning paradigm has shown superiority in PLM-based multi-task learning [5] while TC task naturally consists of multiple tasks. This is the second reason why we leverage the prompt learning paradigm to solve the TC task.
> We found that previous TC methods only leverage subtasks as auxiliary supervision for training representations, which are used only for final prediction. They ignore the potential of prediction results of subtasks to improve the TC task while recent studies have found that the step-by-step prediction method performs better than directly predicting the final outputs [6]. Thus, **our second contribution is to further improve prompt-based multi-task learning for TC by task collaboration through the design of the result context.**
>
>
> **Q3. The Choice of the Backbone LM**
>
> A3. Our main contributions are not limited to the choice of the backbone LM. The choice of BERT mainly aims to align with the experimental settings of previous works for fair comparison.
> We appreciate the interesting idea of leveraging causal LMs or LLMs to perform the TC task, which is inspirational for our future work.
>
> Thank you for the valuable suggestions. Please let us know if you have any other comments/questions.
>
> Best,
>
> Authors
>
> **References**
>
> [1] Cai X, Xu H, Xu S, et al. Badprompt: Backdoor attacks on continuous prompts[J]. Advances in Neural Information Processing Systems, 2022, 35: 37068-37080.
>
> [2] Xu S, Pang L, Shen H, et al. Match-Prompt: Improving Multi-task Generalization Ability for Neural Text Matching via Prompt Learning[C]//Proceedings of the 31st ACM International Conference on Information & Knowledge Management. 2022: 2290-2300.
>
> [3] Chen X, Li L, Zhang N, et al. Decoupling knowledge from memorization: Retrieval-augmented prompt learning[J]. Advances in Neural Information Processing Systems, 2022, 35: 23908-23922.
>
> [4] Brown T, Mann B, Ryder N, et al. Language models are few-shot learners[J]. Advances in neural information processing systems, 2020, 33: 1877-1901.
>
> [5] Fu J, Ng S K, Liu P. Polyglot Prompt: Multilingual Multitask Prompt Training[C]//Proceedings of the 2022 Conference on Empirical Methods in Natural Language Processing. 2022: 9919-9935.
>
> [6] Wei J, Wang X, Schuurmans D, et al. Chain-of-thought prompting elicits reasoning in large language models[J]. Advances in Neural Information Processing Systems, 2022, 35: 24824-24837.
>
> [7] Arous I, Dolamic L, Cudré-Mauroux P. TaxoComplete: Self-Supervised Taxonomy Completion Leveraging Position-Enhanced Semantic Matching[C]//Proceedings of the ACM Web Conference 2023. 2023: 2509-2518.

---

### Official Review · Reviewer_j9Kp · 2023-08-11

**Soundness:** 3

**Excitement:**

2: Mediocre: This paper makes marginal contributions (vs non-contemporaneous work), so I would rather not see it in the conference.

**Missing References:**

@inproceedings{xu2022taxoprompt,

  title={TaxoPrompt: A Prompt-based Generation Method with Taxonomic Context for Self-Supervised Taxonomy Expansion.},

  author={Xu, Hongyuan and Chen, Yunong and Liu, Zichen and Wen, Yanlong and Yuan, Xiaojie},

  booktitle={IJCAI},

  pages={4432--4438},

  year={2022}
}

**Paper Topic And Main Contributions:**

The paper introduces a multi-task prompt learning method for tackling the problem of automatic taxonomy completion. This approach is characterized by its collaboration between two subtasks, specifically attaching concepts to the hypernym and hyponym within existing taxonomies. The proposed method formulates the completion task in a masked language model form, enabling the system to learn how to attach non-leaf concepts with limited labeled data. By integrating the results of subtasks in a contextual manner, the system allows collaborative completion. The paper also provides experimental evidence to demonstrate the superiority of TacoPrompt over existing methods on three benchmark datasets.

**Questions For The Authors:**

According to Table 2, it’s interesting that two versions of the proposed method are good at the different subtasks. Specifically, TacoPrompt-Hidden is better at leaf attachment while TacoPrompt-Hidden is more suitable for Non-leaf attachment. Is there any analysis on this question?

**Reasons To Accept:**

(1) Interesting Method: The proposed approach formulates the task as a multi-task prompt learning problem, which seems a interesting perspective for taxonomy completion.

(2) Robust Experimental Validation: The paper conducts comprehensive experiments to study diffierent aspects of the targeted task and the proposed method based on three benchmark datasets.

(3) Clear Writing: The paper is well-written and concise, allowing for easy understanding of the main concepts and contributions of the paper.

**Reasons To Reject:**

(1) Missing Important Reference: This paper misses the most related reference from IJCAI22, titled “TaxoPrompt: A Prompt-based Generation Method with Taxonomic Context for Self-Supervised Taxonomy Expansion”. This missed reference is the first to propose the prompt based method to solve the similar task (although the transcript under the review claims itself is the first one).

(2) Unconvincing Results: According to Table 2, the proposed model greatly exceeds the baseline in different evaluation indicators in the three datasets. For example, the strongest baseline, QEN, 25% is exceeded on Hits@10 on the SemEval-Food dataset. But from the design of the method, I don't see enough advantages to support such an obvious improvement. Additionally, according to the ablation experiments in Table 3, the simplest base models in this paper, FT-ST, and the strongest baseline, QEN, use the same backbone LM (Bert). But there is a significant gap between them in the experimental results. Specifically, FT-ST is 19% higher than QEN on R@10 on the Food dataset. I think this result needs more explanation to make it convincing.

**Reproducibility:**

3: Could reproduce the results with some difficulty. The settings of parameters are underspecified or subjectively determined; the training/evaluation data are not widely available.

**Reviewer Confidence:**

3: Pretty sure, but there's a chance I missed something. Although I have a good feel for this area in general, I did not carefully check the paper's details, e.g., the math, experimental design, or novelty.

---

> ### Author Rebuttal · Authors · 2023-08-28
>
> Thank you for the thoughtful comments and valuable suggestions. Below, we provide our response to the questions and concerns.
>
> **Q1. Missing Reference: IJCAI 2022 TaxoPrompt**
>
> A1. We must argue that the taxonomy completion (TC) task is more than a ‘similar task’ to the taxonomy expansion (TE) task.
> First, the task input is different. TC inputs candidate pairs of <parent, child> for each query whereas TE only inputs a single candidate parent.
> This input difference brings the non-leaf attachment challenge to TC: the TC task may degenerate into the TE task with limited non-leaf training samples [1].
> Second, TC naturally consists of multiple tasks compared to composed to TE, which only consists of a single task. Thus, only TC is concerned with better multi-task learning methods.
>
> We propose a prompt-based multi-task learning framework to address TC since the prompt learning paradigm has been proven effective in learning with few labelled training samples [2] and multi-task learning [3]. To further improve prompt-based multi-task learning for TC, we design the result context to consider the results of subtasks in the final prediction.
> **Overall**, we aim to address research problems that only exist in TC and that’s why we focus on introducing the related studies of TC instead of TE.
>
> TaxoPrompt is proposed to address the TE task by generating the candidate hypernym of the query concept through the prompt learning paradigm. However, it cannot generate a pair of candidate hypernym and hyponym to address the TC task. Hence, we claimed that we propose the first prompt learning-based framework for the TC task.
>
> **Q2. Unconvincing Results**
>
> A2. The base models **utilize the same backbone LM in different ways**. Our simplest base model FT-ST feeds description sentences of the query, parent and child as a triplet to the backbone LM to directly perform TC. The base models of previous TC methods (including the strongest baseline QEN) input individual description sentences into the backbone LM to derive node-level embeddings of the query, parent and child, which are concatenated and then decoded for TC. The experimental results show that our base model is more effective than those in previous TC methods.
>
> **However, that doesn't mean our contributions mainly come from such a stronger base model.** According to Table 3 in the paper, the effectiveness of our proposed framework on claimed (1) better non-leaf attachment by addressing TC with prompt-based learning paradigm, and (2) task collaboration by the result context to improve prompt-based multi-task learning for TC is demonstrated. For example, our framework averagely increases R@10 of non-leaf queries by 11.65% over FT-ST on the Food dataset.
>
> **Besides, we supplement the reproduction results of the latest published work TaxoComplete [4]**, which could be seen as contemporaneous work to our submission according to the Paper Submission Information of EMNLP 2023. I hope these supplementary results can provide more convincing comparisons and address your concerns. Note that  '#' and '*' refer to leaf and non-leaf queries, respectively.
>
>
> | Food      | MR↓  | MRR  | R@1  | R@5  | R@10 | H@1  | H@5  | H@10 | MRR# | H#5  | R#10 | MRR* | H*5  | R*10 |
> | ------------ | ---- | ---- | ---- | ---- | ---- | ---- | ---- | ---- | ---- | ----  | ---- | ---- | ---- | ---- |
> | TaxoComplete | 96.039  | 0.507 | 13.9 | 30.0 | 38.3 | 28.4 | 60.1 | 71.6 | 0.764 | 65.6 | 70.9 | 0.322 | 34.6 | 14.8 |
> | Ours-Chain  | 51.643 | 0.717 | 28.6 | 51.4 | 60.8 | 60.1 | 85.8 | 89.2 | 0.886 | 87.8 | 86.7 | 0.587 | 76.0 | 40.9 |
> | Ours-Hidden | 47.423 | 0.708 | 30.9 | 51.1 | 60.1 | 64.9 | 85.8 | 86.5 | 0.899 | 87.8 | 87.4 | 0.561 | 76.0 | 39.2 |
>
> | MeSH     | MR↓  | MRR  | R@1  | R@5  | R@10 | H@1  | H@5  | H@10 | MRR# | H#5  | R#10 | MRR* | H*5  | R*10 |
> | ------------ | ---- | ---- | ---- | ---- | ---- | ---- | ---- | ---- | ---- | ---- | ---- | ---- | ---- | ---- |
> | TaxoComplete | 93.741 | 0.565 | 10.7 | 30.5 | 43.6 | 20.6 | 50.2 | 64.3 | 0.616 | 49.4 | 51.5 | 0.526 | 52.0 | 37.5 |
> | Ours-Chain |  76.579 | 0.678 | 17.4 | 41.8 | 56.4 | 39.6 | 72.9 | 82.4 | 0.833 | 77.4 | 78.2 | 0.583 | 63.2 | 43.0 |
> | Ours-Hidden |  49.140 | 0.674 | 17.9 | 42.4 | 55.9 | 40.7 | 74.6 | 84.6 | 0.868 | 79.0 | 81.5 | 0.554 | 65.1 | 40.2 |
>
> | Verb     | MR↓  | MRR  | R@1  | R@5  | R@10 | H@1  | H@5  | H@10 | MRR# | H#5  | R#10 | MRR* | H*5  | R*10 |
> | ------------ | ---- | ---- | ---- | ---- | ---- | ---- | ---- | ---- | ---- | ---- | ---- | ---- | ---- | ---- |
> | TaxoComplete | 596.347 | 0.424 | 7.7 | 21.1 | 30.4 | 12.2 | 31.6 | 42.6 | 0.529 | 33.2 | 44.4 | 0.319 | 25.4 | 16.4 |
> | Ours-Chain   | 597.098 | 0.546 | 18.5 | 35.5 | 45.2 | 28.3 | 49.3 | 58.9 | 0.692 | 52.9 | 61.9 | 0.385 | 34.7 | 26.8 |
> | Ours-Hidden  |  436.799 | 0.557 | 18.3 | 36.9 | 46.5 | 28.0 | 52.3 | 62.5 | 0.762 | 56.5 | 65.8 | 0.370 | 35.2 | 25.3 |
>
>
> **Q3. Analysis of Two Variants**
>
> A3. The results of subtasks are represented to the LM as explicit ‘yes’ or ‘no’ tokens and hidden vectors respectively in TacoPrompt-Chain and in TacoPrompt-Hidden. The difference in how LM understands these two result representations could lead to respective advantages of these two variants in non-leaf and leaf attachment scenarios.
>
> Thank you for the valuable suggestions. Please let us know if you have any other comments/questions.
>
> Best,
>
> Authors
>
> **References**
>
> [1] Wang S, Zhao R, Zheng Y, et al. Qen: Applicable taxonomy completion via evaluating full taxonomic relations[C]//Proceedings of the ACM Web Conference 2022. 2022: 1008-1017.
>
> [2] Brown T, Mann B, Ryder N, et al. Language models are few-shot learners[J]. Advances in neural information processing systems, 2020, 33: 1877-1901.
>
> [3] Fu J, Ng S K, Liu P. Polyglot Prompt: Multilingual Multitask Prompt Training[C]//Proceedings of the 2022 Conference on Empirical Methods in Natural Language Processing. 2022: 9919-9935.
>
> [4] Arous I, Dolamic L, Cudré-Mauroux P. TaxoComplete: Self-Supervised Taxonomy Completion Leveraging Position-Enhanced Semantic Matching[C]//Proceedings of the ACM Web Conference 2023. 2023: 2509-2518.

---

### Meta-Review · Area_Chair_UK4d · 2023-09-18

**Recommendation:** 4

**Metareview:**

The reviewers express mixed views towards the paper that proposes a new prompt-based framework for taxonomy completion.

Collectively, the strengths of the paper include a sound approach to the taxonomy completion task using prompt learning. Reviewers appreciated the extensive experiments on benchmark taxonomy datasets, the presentation of two innovative variants of TacoPrompt, and the solid empirical results, with significant improvement over previous methods.

However, questions have been raised about the novelty of using the Masked Language Model (MLM) and prompts in the taxonomy completion task, as the MLM/generation formulation has been studied for other IE tasks. Also, further comparisons with other techniques is required for a comprehensive evaluation

---

### Decision · Program_Chairs · 2023-10-07

**Decision:**

Accept-Main

**Comment:**

The reviewers express mixed views towards the paper that proposes a new prompt-based framework for taxonomy completion.

Collectively, the strengths of the paper include a sound approach to the taxonomy completion task using prompt learning. Reviewers appreciated the extensive experiments on benchmark taxonomy datasets, the presentation of two innovative variants of TacoPrompt, and the solid empirical results, with significant improvement over previous methods.

However, questions have been raised about the novelty of using the Masked Language Model (MLM) and prompts in the taxonomy completion task, as the MLM/generation formulation has been studied for other IE tasks. Also, further comparisons with other techniques is required for a comprehensive evaluation